# RashomonGB: Analyzing the Rashomon Effect and Mitigating Predictive Multiplicity in Gradient Boosting

**Hsiang Hsu[1], Ivan Brugere[2], Shubham Sharma[2], Freddy Lecue[2], and Chun-Fu Chen[1]**

[1]JPMorganChase Global Technology Applied Research
[2]JPMorganChase AI Research
{hsiang.hsu, ivan.brugere, shubham.x2.sharma}@jpmchase.com
{freddy.lecue, richard.cf.chen}@jpmchase.com

## Abstract

The Rashomon effect is a mixed blessing in responsible machine learning. It enhances the prospects of finding models that perform well in accuracy while adhering to ethical standards, such as fairness or interpretability. Conversely, it poses a risk to the credibility of machine decisions through predictive multiplicity. While recent studies have explored the Rashomon effect across various machine learning algorithms, its impact on gradient boosting—an algorithm widely applied to tabular datasets—remains unclear. This paper addresses this gap by systematically analyzing the Rashomon effect and predictive multiplicity in gradient boosting algorithms. We provide rigorous theoretical derivations to examine the Rashomon effect in the context of gradient boosting and offer an information-theoretic characterization of the Rashomon set. Additionally, we introduce a novel inference technique called RashomonGB to efficiently inspect the Rashomon effect in practice. On more than 20 datasets, our empirical results show that RashomonGB outperforms existing baselines in terms of improving the estimation of predictive multiplicity metrics and model selection with group fairness constraints. Lastly, we propose a framework to mitigate predictive multiplicity in gradient boosting and empirically demonstrate its effectiveness.

## 1 Introduction

Large-scale, complex data and the pursuit of superior performance in machine learning (ML) models have led to increased complexity in both the models themselves and the training algorithms [44]. As a result, it is more likely to find a plethora of distinct models, such as those found in local minima, that exhibit statistically indistinguishable performance (e.g., test accuracy) [23]. This phenomenon, known as the *Rashomon effect* [12], has urged researchers to reconsider its impact on ML models when deployed in real-world scenarios [22, 10, 30, 26].

The impacts of the Rashomon effect reveals two sides of the same coin in responsible ML. On one hand, it benefits the current trend of developing algorithms that prioritize responsible ML principles beyond merely optimizing for accuracy. These principles often include interpretability [64], causality [37], group fairness [20], counterfactual explanations [39], and feature interactions [51]. The abundance of models with competing performance allows compliance with these principles without significant compromises in performance. For instance, algorithmic fairness often faces a trade-off with accuracy; the Rashomon effect allows a fairness intervention algorithm to identify a more optimal balance among models with statistically similar performance [20]. On the other hand, the Rashomon effect presents a risk to the credibility of machine decisions known as *predictive multiplicity* [56], where competing models, generated by simply varying randomness[1] in the training processes, yield

---

[1]Such randomness includes different seeds, weight initialization, splits of mini-batches, etc.

conflicting predictions for some individual samples. If left unaddressed, conflicting predictions can lead to discrimination and unfairness, hidden under the guise of algorithmic randomness. This can adversely affect certain individuals without revealing significant statistical differences from non-discriminatory models [22]. The negative societal impacts of predictive multiplicity and inconsistent decisions have been recently studied under various frameworks[2] such as prediction uncertainty [35, 2], predictive churn [59, 75], and predictive multiplicity [56, 67, 9, 42].

To understand the Rashomon effect and address predictive multiplicity, recent studies have focused on characterizing competing models and efficiently searching for them across different ML models. For instance, predictive multiplicity caused by linear classifiers can be computed via mixed integer programming [56, 74]. Competing models from sparse decision trees are exactly characterized and searched by sub-tree pruning [76]. In special cases such as ridge regression and generalized additive models, the forms of the set of competing models are analytically derived [67, 16]. More recently, test-time dropout has been utilized to search for competing models for neural networks [43].

In this paper, we focus on another ML algorithm, *gradient boosting* [65], which is widely applied to tabular datasets [36]. Gradient boosting differs fundamentally from other ML algorithms in its sequential approach: rather than training a model as a single entity, gradient boosting breaks down the training process into a sequence of sub-learning problems. This sequential training pipeline not only facilitates the analysis of the Rashomon effect but also offers new methodologies for model selection and reducing predictive multiplicity. To the best of our knowledge, our work is the first to explore the Rashomon effect for gradient boosting. The main contributions of this work include:

1. We formalize and investigate the Rashomon effect induced by gradient boosting, employing statistical learning and information theory to analyze its behavior (Section 3). Specifically, we leverage information-theoretic measures to characterize the impact of dataset properties on the Rashomon effect.

2. We introduce `RashomonGB`, an efficient method that explores an exponential search space versus baseline methods which search linearly. (Section 3.1).

3. We implement `RashomonGB` on several real-world (large-scale) tabular and image datasets, and empirically demonstrate the competing models obtained with `RashomonGB` can greatly improve the estimation of predictive multiplicity (Section 4.1), and model selection with additional responsible ML principles such as group fairness (Section 4.2).

4. We propose two methods to mitigate predictive multiplicity for gradient boosting and experimentally validate the methods on 18 tabular datasets (Section 4.3).

Omitted proofs, additional explanations and discussions, details on experiment setups and training, and additional experiments are included in the Appendix. Code to reproduce our experiments can be accessed at https://github.com/jpmorganchase/Rashomon-gradient-boosting.

## 2  Background and related work

Consider a dataset $\mathcal{S} = \{\mathbf{s}_i\}_{i=1}^n$ drawn i.i.d. from $P_S$, where each $\mathbf{s}_i$ is a pair $(\mathbf{x}_i, y_i)$ consisting of a feature vector $\mathbf{x}_i = [\mathbf{x}_{i1}, \cdots, \mathbf{x}_{id}]^\top \in \mathcal{X} \subseteq \mathbb{R}^d$ and a target $y_i \in \mathbb{R}$. Let $X$ and $Y$ be the random variables for the feature $\mathbf{x}_i$ and target $y_i$ respectively, and $S = X \times Y$. We denote by $\mathcal{H}$ a hypothesis space of functions that map from $\mathcal{X}$ to $\mathcal{Y}$. The loss function used to evaluate model performance is denoted by $\ell : \mathcal{H} \times \mathcal{S} \to \mathbb{R}^+$ and $L_{P_S}(h) \triangleq \mathbb{E}_{P_S}[\ell(h, S)]$ the population risk. As usual, the population risk is approximated by the empirical risk $L_{\mathcal{S}}(h) \triangleq \frac{1}{n} \sum_{i=1}^n \ell(h, \mathbf{s}_i)$. We denote the empirical risk minimizer as $h^* = \operatorname{argmin}_{h \in \mathcal{H}} L_{\mathcal{S}}(h) \in \mathcal{H}$. We denote $\nabla_x v(x)$ the gradient of $v(x)$ w.r.t. $x$, and $\mathbb{1}[\cdot]$ the indicator function. For two random variables $X$ and $Y$, the mutual information between them is defined as $I(X; Y) = D_{\mathsf{KL}}(P_{X,Y} \| P_X P_Y)$ [21], where $D_{\mathsf{KL}}(P \| Q) = \mathbb{E}_P[\log P/Q]$ is the Kullback-Leibler (KL) divergence [47]. Finally, we let $[k] \triangleq [1, \cdots, k]$.

**The Rashomon sets and exploring models therein.**  The studies on the Rashomon effect typically start with searching for models in the *Rashomon set* [67], the set of all models in the hypothesis space

---

[2]Both prediction uncertainty and predictive multiplicity consider the arbitrariness of machine outputs, with predictive multiplicity specifically addressing models with competing performance. Predictive churn, on the other hand, focuses on the instability of decisions before and after updating models with new data.

$\mathcal{H}$ whose population risks are comparable to that of a given model[3] $h^* \in \mathcal{H}$, i.e.,

$$\mathcal{R}(\mathcal{H}, \mathcal{S}, h^*, \epsilon) \triangleq \{h \in \mathcal{H}; L_{P_S}(h) \leq L_{P_S}(h^*) + \epsilon\}, \tag{1}$$

where $\epsilon \geq 0$ is a Rashomon parameter that determines the size of the Rashomon set. However, when the hypothesis space $\mathcal{H}$ is large (e.g., neural network architectures, tree ensembles, etc.), exhaustively identifying all models within the Rashomon set becomes computationally infeasible. Therefore, it is customary to approximate the full Rashomon set by a subset with $m$ models called an *empirical Rashomon set*, $\mathcal{R}^m(\mathcal{H}, \mathcal{S}, h^*, \epsilon) \triangleq \{h_1, \cdots, h_m \in \mathcal{H}; h_i \in \mathcal{R}(\mathcal{H}, \mathcal{S}, h^*, \epsilon), \ \forall i \in [m]\}$. In practice, the $m$ models in the empirical Rashomon set are mainly obtained by `re-training` (See Appendix B.2 for more discussions). The `re-training` strategy re-trains models with different random initializations and rejects those that disobey the loss deviation constraint in Eq. (1) until $m$ models are collected [67, 48]. However, re-training models repeatedly is time-consuming with large datasets or complex architectures. To improve efficiency, a recent strategy involves collecting distinct models with competing performance during the inference phase. For example, Hsu et al. [43] proposed an *inference-time dropout* strategy for convolutional neural networks (CNNs). Despite these advances, exploring Rashomon sets for ensemble learning methods like boosting remains unexplored.

**Predictive multiplicity.**    Predictive multiplicity undermines the credibility of decisions made by ML algorithms, making its measurement an active research area. Predictive multiplicity metrics can be categorized based on whether they are defined on output decisions (i.e., thresholded predictions/scores after $\mathrm{argmax}$) or on output scores in the probability simplex. For example, *ambiguity* and *discrepancy*, measure the proportion of samples with conflicting decisions from models within the Rashomon set [56]. *Disagreement*, in a similar flavor, assess the probability of conflicting decisions per sample [48]. In contrast, score-based metrics estimate various aspects such as score variance/std. [52, 17, 9], the viable range of scores (termed *viable prediction range (VPR)* by Watson-Daniels et al. [74])), or score spread in the probability simplex (referred to as *Rashomon Capacity (RC)* by Hsu and Calmon [42]). Due to space constraints, we defer the mathematical formulations of these predictive multiplicity metrics and discussions on estimating these metrics with the empirical Rashomon sets to Appendix B.2. These predictive multiplicity metrics are often estimated using the empirical Rashomon set $\mathcal{R}^m(\mathcal{H}, \mathcal{S}, h^*, \epsilon)$. As $m$ increases, the empirical Rashomon set better approximates the Rashomon set, leading to more precise estimations of predictive multiplicity metrics.

**Mitigating predictive multiplicity.**    Mitigating predictive multiplicity ensures that decisions made by ML algorithms are consistent. The main strategy for this is to combine decisions from competing models. Roth et al. [63] reconcile conflicting decisions from two different models in the Rashomon set to improve the disagreement in predictions. Combining decisions from multiple models falls under the umbrella of model averaging in ensemble learning [9, 42, 52]. Model averaging is a special ensemble learning that collects multiple base models, often referred to as weak learners, and combines them *in parallel*. As averaging model outputs reduces the variance, it is a natural choice for diminishing predictive multiplicity and has been reported in several studies. For instance, Black et al. [9] proposed a selective averaging that leverages certifiably-robust predictions to mitigate the problem of inconsistency with a probabilistic guarantee. Hsu and Calmon [42, Section A.4.5] observe that random forest classifiers exhibit a lower Rashomon Capacity compared to decision tree classifiers. Furthermore, Long et al. [52] demonstrate that the probability of significant deviated predictions in model averaging diminishes exponentially with the number of models in the average.

Ensemble learning is not limited to parallel combinations. Another popular branch involves combining models *sequentially*, known as boosting algorithms. Despite the widespread use of boosting algorithms and their superior performance over neural networks on tabular datasets [36], boosting algorithms have been mostly overlooked in the literature on the Rashomon effect and predictive multiplicity. This paper aims to address this gap, as outlined in the next section. The works most closely related to ours involve prediction uncertainty estimation in gradient boosting, such as NGBoost [27, 55], PGBM [70], and IBUG [13], which consider probabilistic predictions from regression trees in Bayesian settings. However, these studies do not frame the concept of arbitrariness in predictions within the context of the Rashomon effect and may overestimate arbitrariness with models that do not have similar performance. To the best of our knowledge, this work is the first that investigates the impact of the Rashomon effect and predictive multiplicity on gradient boosting.

---

[3]It is common to choose the model as an empirical risk minimizer.

## 3 Analyzing the Rashomon effect in gradient boosting

We begin with a brief introduction to gradient boosting and discuss how its sequential combination approach can inspire a new method for finding competing models in the Rashomon set. We then provide a high-probability bound on the Rashomon set using information theory, making the first connection between the size of the Rashomon set and data quality as measured by mutual information. The proofs of the propositions in this section are included in Appendix A.

Boosting algorithms select a sequence of weak learners[4], $h_0, \cdots, h_T \in \mathcal{H}$, such that the additive expansion $f_T(\mathbf{x}) = \sum_{t=0}^{T} \alpha h_t(\mathbf{x}) \in \mathcal{F}$ with $\alpha > 0$ minimizes the empirical risk $L_{\mathcal{S}}(f_T)$ [34, 57, 66, 19]. Different choices of $\mathcal{H}$ and the method for selecting $h_t \in \mathcal{H}$ have led to various boosting algorithms (see Appendix B.1). Here, our focus lies on gradient boosting [66, 34], which starts with a constant model $h_0(\mathbf{x}) = \operatorname{argmin}_{h_0 \in \mathbb{R}} \sum_{i=1}^{n} \ell(h_0, \mathbf{s}_i)$, and iteratively extends the model $f_t(\mathbf{x}_i)$ with

$$f_t(\mathbf{x}) = f_{t-1}(\mathbf{x}) + \operatorname*{argmin}_{h_t \in \mathcal{H}} \| - \nabla L_{\mathcal{S}}(f_{t-1}) - h_t \|_2^2 = f_{t-1}(\mathbf{x}) + \operatorname*{argmin}_{h_t \in \mathcal{H}} \sum_{i=1}^{n} \| h_t(\mathbf{x}_i) - r_{ti} \|_2^2,$$

(2)

where $r_{ti} = - \left[ \frac{\partial \ell(f_{t-1}, \mathbf{s}_i)}{\partial f_{t-1}} \right]$ is the pseudo-residual of sample $i$ from weaker learner $h_t$. Indeed, for a regression problem with a mean squared error (MSE) loss function, $h_0^* = \sum_{i=1}^{n} y_i / n$ and $r_{ti} = 2(y_i - f_{t-1}(\mathbf{x}_i))$—the part of the $y_i$ that cannot be explained by the current model $f_{t-1}(\mathbf{x}_i)$. For a binary classification problem with a cross-entropy (CE) loss, $h_0^* = \log \sum_{i=1}^{n} \mathbb{1}[y_i = 1] / \sum_{i=1}^{n} \mathbb{1}[y_i = 0]$, and can be understood as regression on the log-likelihoods. We set $\alpha = 1$ in this section for the sake of theoretical analysis. For the convergence and consistency analysis of gradient boosting, see, e.g., Zhang and Yu [79] and Telgarsky [71].

### 3.1 Building Rashomon sets for gradient boosting

The iterative training procedure of gradient boosting in Eq. (2) allows us to convert the original learning problem $\min_{f \in \mathcal{F}} L_{\mathcal{S}}(f)$ into a sequence of learning problems $\min_{h_t \in \mathcal{H}} L_{\mathcal{S}_t}(h_t)$, for $t \in [1, 2, \cdots, T]$, where $\mathcal{S}_t = \{(\mathbf{x}_i, r_{ti})\}_{i=1}^{n}$. In other words, the weak learner $h_t$ aims to fit the pseudo-residuals in each iteration, and inducing a *residual* Rashomon set (cf. Figure 1),

$$\mathcal{R}_t(\mathcal{H}, \mathcal{S}_t, h_t^*, \epsilon) \triangleq \{ h_t \in \mathcal{H}; L_{P_{\mathcal{S}_t}}(h_t) \leq L_{P_{\mathcal{S}_t}}(h_t^*) + \epsilon \},$$

(3)

where $h_t^*$ is any given model such as the empirical risk minimizer in each iteration. Eq. (3) suggests an alternative to building the entire Rashomon set $\mathcal{R}(\mathcal{F}, \mathcal{S}, f_T^*, T\epsilon)$ by iteratively building the residual Rashomon sets for each iteration, i.e.

$$\mathcal{R}_1(\mathcal{H}, \mathcal{S}_1, h_1^*, \epsilon) \times \cdots \times \mathcal{R}_t(\mathcal{H}, \mathcal{S}_t, h_t^*, \epsilon) \times \cdots \times \mathcal{R}_T(\mathcal{H}, \mathcal{S}_T, h_T^*, \epsilon) \supseteq \mathcal{R}(\mathcal{F}, \mathcal{S}, f_T^*, T\epsilon). \quad (4)$$

Eq. (4) comes from the fact that, for classification tasks, gradient boosting actually performs a regression on the log-likelihood using the MSE loss. The pseudo-residual of the MSE loss exhibits a linear relationship between the prediction and the output of each iteration, allows us to aggregate the losses across iterations. Equation (4) will be clarified further in Proposition 2.

In practice, if we perform $m$ re-training in each iteration, and obtain the empirical residual Rashomon set $\mathcal{R}_t^m(\mathcal{H}, \mathcal{S}_t, h_t^*, \epsilon)$, the model $f_T(\mathbf{x})$ can be expressed as $f_T(\mathbf{x}) = h_0(\mathbf{x}) + \sum_{t=1}^{T} h_t(\mathbf{x})$, $\forall h_t \in \mathcal{R}_t^m(\mathcal{H}, \mathcal{S}_t, h_t^*, \epsilon)$. Since in each of the $T$ iterations, there are $m$ candidate models in $\mathcal{R}_t^m(\mathcal{H}, \mathcal{S}_t, h_t^*, \epsilon)$, there are a total of $m^T$ possibilities for $f_T(\mathbf{x}) \in \mathcal{R}(\mathcal{F}, \mathcal{S}, f_T^*, T\epsilon)$ (see Appendix B.3 for visualization). We term the process of building the empirical residual Rashomon sets for the entire Rashomon set $\mathcal{R}(\mathcal{F}, \mathcal{S}, f_T^*, T\epsilon)$ as *Rashomon gradient boosting* or `RashomonGB` in short. `RashomonGB` can be straightforwardly implemented by training $m$ models (i.e., weak learners) at each iteration that meet the loss constraints defined by the Rashomon set. This is fundamentally different from simply performing $m$ re-training of gradient boosting to solve the original learning problem $\min_{f \in \mathcal{F}} L_{\mathcal{S}}(f)$, since the $m$ models at each iteration obtained by `RashomonGB` share the same residual from the previous iteration. During the inference phase `RashomonGB` carries a significant benefit over simply performing $m$ re-training of gradient boosting—both re-training of gradient boosting and `RashomonGB` requires $m \times T$ training when considering all the iterations.

---

[4]The weak learners in gradient boosting are usually decision tree regressors. However, the choice of the weak learners can be generalized to more complex families of functions [49] such as neural networks with [25] or without regularizations [6, GrowNet], and Gaussian processes [69].

However, `RashomonGB`, using a different way to output the predictions at the inference phase, yields exponentially many more models with the same training cost. So far, we have informally introduced `RashomonGB`. In the rest of this section, we provide a rigorous analysis to show the effectiveness of `RashomonGB` by characterizing the Rashomon sets of gradient boosting (or more generally, the Rashomon sets of the iterative training procedure). The analysis collectively ensure that `RashomonGB` is not only practical but also robust, enhancing its applicability in various real-world scenarios.

### 3.2 Information-theoretic characterization of the Rashomon set

Characterizing the entire Rashomon set, given a learning problem and dataset, has posed as a significant computational challenge (refer to Section 2). Specifically, for gradient boosting, delineating the Rashomon set $\mathcal{R}(\mathcal{F}, \mathcal{S}, f_T^*, \epsilon)$ entails identifying all models $f_T \in \mathcal{F}$ such that $L_{P_S}(f_T) \leq L_{P_S}(f_T^*) + \epsilon$ for a given model $f_T^*$. However, Eq. (4) presents a novel approach to approximating the Rashomon set by decomposing it into residual Rashomon sets concerning the weak learners—this method offers valuable insights into analyzing the Rashomon effect in gradient boosting.

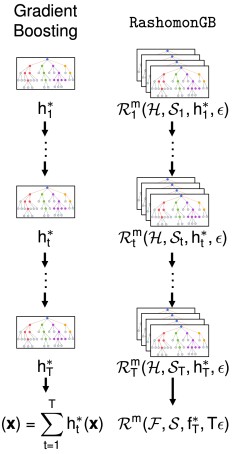

Figure 1: Gradient boosting and `RashomonGB` with $T$ iterations.

Existing literature explores how various hypothesis spaces influence the Rashomon set but frequently neglects the impact of datasets, despite the Rashomon set being inherently tied to the dataset itself (cf. Eq. (1)). In this context, we derive a novel bound on the size of the Rashomon set by expanding the scope of the Rashomon set beyond mere consideration of hypothesis spaces, incorporating the influence of datasets through the lens of statistical learning and information theory. Particularly in the analysis of information-theoretic generalization error bounds [77], a learning algorithm $A(\mathcal{S})$ is conceptualized as a random variable $H$ that generates models within the hypothesis space $\mathcal{H}$. Moreover, the loss function $\ell(h, S)$ is also treated as a random variable, and is further assumed to be $\sigma$-sub-Gaussian[5] [73], a property that effectively generalizes the boundedness assumption of the loss function in information-theoretical analysis. Sub-Gaussianity is a practical property since for a bounded loss function $\ell \in [a, b]$, which can be readily satisfied by clipping the loss[6], $\ell(h, S)$ is guaranteed to be $(b - a)/2$-sub-Gaussian. With these properties in place, the mutual information $I(S; H)$ between $H$ and the random variable of dataset $S$ emerges as a pivotal metric for assessing generalizability. By establishing a connection between generalization error bounds and the definition of a Rashomon set in Eq. (1), and leveraging the properties of a sub-Gaussian loss, we derive the following high-probability bound for a Rashomon set.

**Proposition 1.** *For a dataset $\mathcal{S}$, given an empirical risk minimizer $h^* = \mathrm{argmin}_{h \in \mathcal{H}} L_{\mathcal{S}}(h)$ and a $\sigma$-sub-Gaussian loss $\ell$, with probability at least $1 - \rho$, we have*

$$h \in \mathcal{R}\left(\mathcal{H}, \mathcal{S}, h^*, \sqrt{\frac{8\sigma^2}{n}\left(\frac{2I(S; H)}{\rho} + \ln\frac{4}{\rho}\right)} + \sqrt{\frac{\sigma^2}{n}\ln\frac{4}{\rho}}\right). \tag{5}$$

In contrast to existing analyses of the Rashomon set primarily focused on optimization perspectives, Proposition 1 offers a characterization of the size of the Rashomon set in terms of the controllable probability of the Rashomon parameter $\epsilon$. This sheds light on understanding the Rashomon set from a statistical learning standpoint. Furthermore, the mutual information $I(S; H)$ in Eq. (5) quantifies the uncertainty of the learning algorithm with respect to the dataset $\mathcal{S}$. In essence, $I(S; H)$ serves as a metric for the *multiplicity of models*, thereby contributing to predictive multiplicity. By chain rules, we can decompose the mutual information into two components, each delineating a distinct source responsible for inducing multiplicity, i.e.,

$$I(S; H) = I(Y, X; H) = \underbrace{I(X; H)}_{\text{Model Uncertainty}} + \underbrace{I(Y; H|X)}_{\text{Quality of Data}}. \tag{6}$$

When $I(X; H)$ is small, the model derived from the learning algorithm becomes nearly deterministic, indicating minimal uncertainty regarding model selection. Similarly, $I(Y; H|X)$ can be expressed

---

[5]A random variable $U$ is $\sigma$-sub-Gaussian if $\log \mathbb{E}\left[e^{\lambda(U - \mathbb{E}U)}\right] \leq \lambda^2 \sigma^2 / 2$ for all $\lambda \in \mathbb{R}$.

[6]The MSE loss is also qualified as a sub-Gaussian loss function.

as $I(Y; H|X) = g(Y|X) - g(H|Y, X)$, where $g$ denotes the (conditional) Shannon's entropy [21]. The first component, $g(Y|X)$, evaluates the data quality; if the channel from $X$ to $Y$ is devoid of noise, the conditional entropy will be low. Conversely, if the channel is noisy, the conditional entropy will be lower-bounded by the entropy of the noise source.

The mutual information $I(S; H)$ indeed contains intricate details about the dataset, allowing us to incorporate the dataset's influence into depicting a Rashomon set. This is precisely why we opt for information-theoretic tools over Rademacher complexity [68]. While Rademacher complexity offers "data-dependent" generalization bounds, it lacks the ability to provide substantial insights into the dataset itself. For completeness, we include a theoretical analysis of the size of Rashomon sets based on Rademacher complexity, akin to the high probability bound in Proposition 1, in Appendix C.

Now we move back the characterizing the Rashomon set for `RashomonGB`. Recall that for each iteration $t \in [T]$ in gradient boosting, we may define a residual Rashomon set $\mathcal{S}_t$. By plugging $\mathcal{S}_t$ into $\mathcal{S}$ in Proposition 1, and using the decomposition of the entire Rashomon set $\mathcal{R}(\mathcal{F}, \mathcal{S}, f_T^*, T\epsilon)$ in Eq. (4), we are now equipped to re-evaluate the entire Rashomon set of gradient boosting.

**Proposition 2.** *Let $h_t^*$ be the empirical risk minimizer for each boosting iteration $\mathcal{S}_t$, and $f_T^* = \sum_{t=1}^{T} h_t^*$, then for a $\sigma$-sub-Gaussian loss $\ell$, with probability at least $1 - T\rho$, we have*

$$f_T \in \mathcal{R}\left(\mathcal{F}, \mathcal{S}, f_T^*, T\sqrt{\frac{\sigma^2}{n}\ln\frac{4}{\rho}} + \sum_{t=1}^{T}\sqrt{\frac{8\sigma^2}{n}\left(\frac{2I(\mathcal{S}_t; H)}{\rho} + \ln\frac{4}{\rho}\right)}\right). \tag{7}$$

Proposition 2 suggests that the Rashomon set grows with the boosting iterations—it is due to the increased complexity of the overall model $f_t$. Unless the number of samples $n$ goes to infinity, the Rashomon set has a non-zero size.

### 3.3 The Rashomon effect in each iteration of gradient boosting

The information-theoretic analysis presented in the previous section underscores the importance of the mutual information $I(H; Y|X)$ in determining the size of the Rashomon set and hence the severity of predictive multiplicity. However, existing literature lacks an in-depth analysis of the information contained within the pseudo-residuals for gradient boosting algorithms. Let $R_t$ represents the random variable associated with the residuals $\{r_{ti}\}_{i=1}^{n}$. As $t$ grows, fitting $R_t$ with $X$ becomes progressively challenging, akin to the gradient vanishing problem observed in deep learning [31, 40], thereby resulting in larger conditional mutual information with data $X$. We formalize this observation in the following proposition.

**Proposition 3.** *For both the MSE and CE losses, the mutual information between $H$ and the pseudo-residuals conditioned on the features $X$ is non-decreasing with respect to the boosting iteration, i.e., let $0 \le t_1 \le t_2 \le T$, then $I(H; R_{t_1}|X) \le I(H; R_{t_2}|X)$.*

The essence of Proposition 3 lies in the fact that the information contained in $R_{t_2}$ can be understood as a combination of the information in $R_{t_1}$ and the additional information provided by the models fitted to the residuals between iterations $t_1$ and $t_2$. Proposition 3, Proposition 1, and Eq. (6) collectively hint at a *counter-intuitive* observation: the size of the residual Rashomon set could potentially increase with more boosting iterations, i.e., for $0 \le t_1 \le t_2 \le T$, we have $\epsilon_{t_1} \le \epsilon_{t_2}$ and hence $\mathcal{R}_t\left(\mathcal{H}, \mathcal{S}_{t_1}, h_{t_1}^*, \epsilon_{t_1}\right) \subseteq \mathcal{R}_t\left(\mathcal{H}, \mathcal{S}_{t_2}, h_{t_2}^*, \epsilon_{t_2}\right)$. This implies that conducting additional boosting iterations[7] may not only exacerbate over-fitting but also result in a larger Rashomon set and heightened predictive multiplicity. Note that we do not start with the assumption that $\epsilon$ increases with each iteration; rather, this conclusion emerges from Proposition 3. With a constant $\rho$—as defined in Proposition 1—additional iterations result in increased conditional mutual information, which in turn necessitates a larger $\epsilon$.

Figure 2 provides a simulation with a 20-dimensional Gaussian synthetic datasets with 100 samples, trained with gradient boosting for 10 iterations, where each iteration contains $m = 100$ models. Here, we show the conditional entropy[8] $g(R_t|X)$ instead of the mutual information $I(R_t, X; H)$ in

---

[7]A similar phenomenon where additional information impacts the performance of boosting algorithms, has also been noted in previous studies such as Friedman et al. [33] and Long and Servedio [53].

[8]Estimated with local linear regression [18], which has consistently smaller MSE for small datasets.

Eq. (6) as the estimation of the mutual information is in general a hard task [60]. It is clear that the conditional entropy—and consequently the mutual information—increases as the boosting procedure iterates, leading to a larger Rashomon set. This occurs because the Rashomon effect accumulates across the sequential learning problems addressed in each iteration, highlighting the cumulative impact on diversity within the model space. The Ablation study in Appendix E.4 further clarifies the selection of $\epsilon$ through its iterations. Figure E.9 demonstrates that fixing $\epsilon$ while re-training with different random seeds results in a decreasing percentage $\rho$ of models in the Rashomon set. This implies that to maintain a consistent $\rho$, the chosen Rashomon parameter $\epsilon$ must increase.

As a remark, consider training $m$ models per iteration, and construct the overall Rashomon sets using models obtained from the $T_1$-th and $T_2$-th iterations with $T_1 < T_2$. With the same threshold $\epsilon$ for the Rashomon set, let $\rho_1$ and $\rho_2$ represent the probabilities for iterations $T_1$ and $T_2$, respectively, then by Proposition 3, we have $1 - \rho_1 \geq 1 - \rho_2$. the number of models from the $T_1$-th iteration that are included in the Rashomon set with threshold $\epsilon$ will be $m^{T_1} \times (1 - \rho_1)$. Similarly, for the $T_2$-th iteration, the count will be $m^{T_2} \times (1 - \rho_2)$. It is important to note that although $\rho$ decreases, this reduction is linear with respect to the number of iterations (as suggested by the term $1 - T\rho$ in Proposition 2). However, the number of models generated by `RashomonGB` grows exponentially with the number of iterations. Thus, the total number of models in the Rashomon set will still asymptotically increase with the number of iterations.

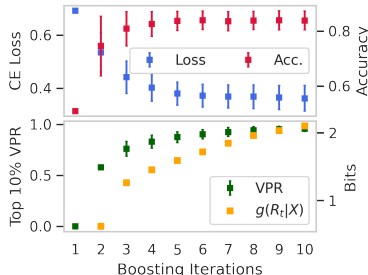

Figure 2: Gradient boosting for binary classification. The conditional entropy of the residuals and the predictive multiplicity (measured by VPR) increase along with the boosting iteration, which matches Proposition 1.

## 4 Applications of Rashomon gradient boosting

We present three use cases demonstrating how `RashomonGB` can be deployed in practice to explore models more effectively in the Rashomon sets: (i) improving the estimation of predictive multiplicity metrics, (ii) fair model selection, and (iii) mitigating predictive multiplicity. Given that there are no existing algorithms specifically designed to explore the Rashomon set for gradient boosting, apart from `re-training` with different seeds [67, 48], we use the `re-training` strategy as our primary baseline for comparison through this section. It's noteworthy that both the `re-training` strategy and `RashomonGB` share the same training complexity. However, during the inference phase, `RashomonGB` exponentially expands the empirical Rashomon set (i.e., with a large $m$ following Section 3.1). In our experiments, we adopt the settings in Friedman [34] with decision tree regressors as weak learners for tabular datasets and CNNs for images. See Appendix D.1 for detailed descriptions and pre-processing of the datasets, Appendix D.2 for detailed training setups, and Appendix E for additional experiments including ablation studies.

### 4.1 Improving the estimation of predictive multiplicity metrics

We estimate the predictive multiplicity metrics (cf. Appendix B.2) on three tabular datasets with binary classes. Two of these datasets are from the financial domain (ACS Income [24] and Credit Card [78]), while the other is from the medical domain (Contraception[5]). The datasets are particularly chosen as predictive multiplicity in these domains could have profound implications for fairness and justice. Note that the ACS Income dataset is an extension of the widely-used UCI Adult dataset [5] with many more samples ($\approx$1.6 million vs. <50k in the UCI Adult), allowing us to compare methods with higher precision. Beyond binary classification and decision-tree weak learners, we use CIFAR-10 [46] as a multi-class case study with CNNs as weak learner, i.e., a setting similar to Badirli et al. [6, GrowNet].

Figure 3 summarizes the estimation of 4 predictive multiplicity metrics using the empirical Rashomon sets obtained from `re-training` and `RashomonGB`. For a more detailed explanation on how to interpret Figure 3, see Appendix E.1. We conduct 10 (and 50 for the CIFAR-10 dataset) `re-training` of gradient boosting with different random seeds; each gradient boosting has $T = 10$ (and $T = 6$ for CIFAR-10) iterations and $m = 10$ (and $m = 50$ for CIFAR-10) models in each iteration. We randomly select 2 out of $m$ models in each boosting iteration and perform `RashomonGB` to obtain

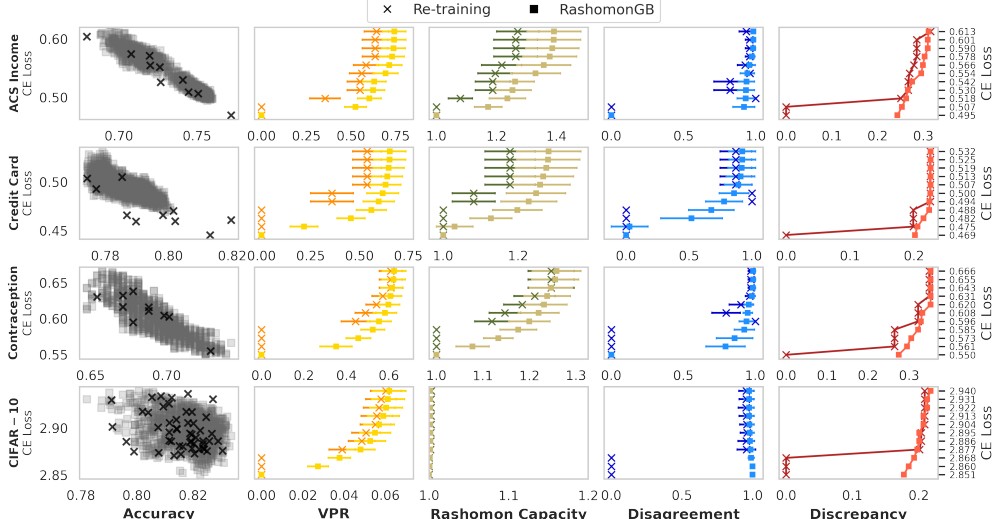

Figure 3: `Re-training` vs. `RashomonGB` in exploring the Rashomon set for predictive multiplicity metrics estimation. In the leftmost column, each marker represents a model. The rightmost 4 figures in a row share the same y-axis for the loss difference (values shown at the right), i.e., $L_{P_S}(h^*) + \epsilon$ in Eq. (1). Higher predictive multiplicity values mean a better estimate. `RashomonGB`, with more models in the Rashomon set, offers more accurate multiplicity estimates under the same loss deviation constraints.

$2^T = 1024$ (and 972 models for CIFAR-10) models[9]. The leftmost column shows the CE loss vs. accuracy for the models obtained by the two methods. It is clear that with the same training cost, `RashomonGB` offers many more models in the Rashomon set that spread wider in the loss-accuracy plane. The rest of the four columns show the estimates of predictive multiplicity metrics given different loss deviation constraints. Since VPR, Rashomon capacity, and disagreement are defined per sample, we plot the mean and std. of the top $10\%$ samples instead[10].

As observed in Figure 3, `RashomonGB` consistently outperforms `re-training` in the ACS Income and Credit Card datasets for all predictive multiplicity metrics. The `RashomonGB` has more advantages especially when `re-training` is incapable of exploring enough models under the loss deviation constraint. For example, in the Contraception dataset, when the loss constraint is under $0.585$, there is only one model obtained from `re-training`, and therefore the corresponding values of predictive multiplicity metrics remain 0. As the loss constraint increases, the Rashomon set from `re-training` has more models and its estimates converge with that of `RashomonGB`. The explanations above indicate that the exploration of diverse models under the same loss constraint largely affects the estimation of predictive multiplicity metrics. For the CIFAR-10 dataset, both VPR and RC values are small, while decision-based metrics, such as disagreement and discrepancy, are large. This implies that despite small score variations, a significant number of samples have scores close to the decision boundary. Consequently, a slight perturbation in scores from a different model in the Rashomon set could lead to a different class after applying $\arg\max$.

We include additional results on three other UCI datasets in Appendix E.2 and a comparison of the computational time[11] to obtained one model from `re-training` and from `RashomonGB` in Appendix E.3. The ablation studies on different types of weak learners $\mathcal{H}$ (e.g., linear regression), depths of decision tree weak learners, the number of boosting iteration $T$ and the number of model in each iteration $m$ are also include in Appendix E.5 to E.8. To elucidate the distinctions between predictive multiplicity and prediction uncertainty estimation in gradient boosting (cf. Section 2), we have compared our re-training strategy, RashomonGB against NGBoost [27, 55], PGBM [70], and IBUG [13] with the UCI Contraception dataset in Appendix E.9.

---

[9]For 3 models per iteration, `RashomonGB` produces $3^{10} \approx 59$k models, exceeding our storage limit.

[10]Predictive multiplicity occurs only on a small portion of samples. The choice of $10\%$ is data-dependent.

[11]For the ACSIncome dataset, the inference time per model is 0.4 seconds for `re-training`, compared to just 0.02 seconds for `RashomonGB`. This indicates that, with the same training cost (54.67 seconds), `RashomonGB` is 20 times more efficient in generating models from the Rashomon set than the `re-training` strategy.

## 4.2 Selecting models with group fairness constraints

We conduct `re-training` and `RashomonGB` on two standard datasets in the algorithmic fairness community, the UCI Adult and COMPAS recidivism datasets [3], with the goal of selecting a model that exhibits better fairness-accuracy trade-offs. We follow the same training procedure and gradient boosting architecture in Section 4.1. We assess the bias across different groups (e.g., female vs. male) by two group-fairness metrics[12], mean equalized odds (MEO) [38] and statistical parity (SP) [29] (cf. Appendix B.4 for details). In both datasets, the group attributes are the binary "race" label. Figure 4 illustrates that with `RashomonGB`, a practitioner has a greater chance to select a model that better satisfies group-fairness constraints without a significant drop in accuracy.

For the UCI Adult dataset, despite that `re-training` achieves the highest accuracy ($\approx 84.5\%$), it violates the MEO with $0.01$. On the other hand, `RashomonGB` provides a model that perfectly complies with fairness constraint (MEO $\approx 0$) with a drop in accuracy less than $1\%$. Similar observations apply to the COMPAS dataset regarding SP. Moreover, we include four additional fairness intervention baselines, `EqOdds` [38], `Rejection` [45], `Reduction` [1], and `FaiRS` [20]. See Appendix B.4 for a brief introduction of the fairness intervention baselines. The most relevant baseline to `RashomonGB` is `FaiRS`, which modifies the `Reduction` approach to address fairness intervention problems across the models in the Rashomon set. `FaiRS` is originally implemented for logistic regression, and we adapt it to gradient boosting.

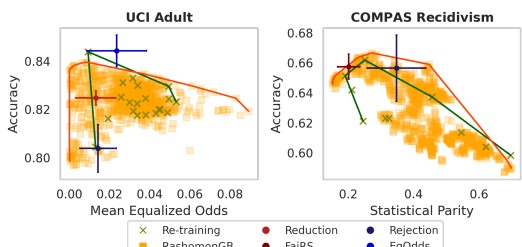

Figure 4: Fairness-accuracy trade-off for `re-training` vs. `RashomonGB` on test set. Each marker represents a model. A better trade-off means a smaller group-fairness metric (MEO or SP) and a higher accuracy, i.e., the top left area. For both datasets, `RashomonGB` provides more better models to select that complies with the fairness constraints whilst having the highest accuracy. For UCI Adult, the CE loss of the models from `RashomonGB` is $0.38 \pm 0.02$ and $0.64 \pm 0.02$ for COMPAS.

For both UCI Adult and COMPAS datasets, `RashomonGB` encompasses nearly all models (excluding `EqOdds`) from the fairness intervention baselines–without explicit fairness intervention–indicating that `RashomonGB` gives a "rich" Rashomon set for model selection with additional fairness considerations. The advantage of RashomonGB becomes even more pronounced when dealing with larger datasets. In such scenarios, re-training and re-training-based fairness intervention algorithms, such as `Reduction` (and `FaiRS`) and `Rejection`, may incur significantly higher training costs. Model selection with `RashomonGB` is not limited the group fairness purposes; this experiment serves as an initial demonstration of the diversity of `RashomonGB` use cases. In Section 4.3 we demonstrate explicit feedback in the model selection process.

## 4.3 Mitigating predictive multiplicity

The `RashomonGB` framework, which involves training $m$ distinct models in each iteration, offers the added benefit of reducing predictive multiplicity in gradient boosting. The $m$ models in the empirical Rashomon set for $\mathcal{R}^m(\mathcal{H}, \mathcal{S}_t, h_t^*, \epsilon)$ for the $t^{th}$ iteration can either be selected (based on the least losses) or aggregated (similarly to model averaging in Section 2). Building on these concepts, we propose two approaches to reduce predictive multiplicity in gradient boosting: (i) *model selection* with reweighted loss (`MS`), and (ii) *intermediate ensembles* during boosting iterations (`IE`).

Let $\ell_{i,j} = \ell(h_j, \mathbf{s}_i)$ be the loss evaluated at sample $\mathbf{s}_i$ for model $h_j \in \mathcal{R}^m(\mathcal{H}, \mathcal{S}_t, h_t^*, \epsilon)$, and let $\bar{\ell}_i = \frac{1}{m} \sum_{j=1}^{m} \ell_{i,j}$ be the mean loss. The `MS` method considers the reweighted loss for each model in the empirical Rashomon set using $\ell_{h_j} \triangleq \sum_{i=1}^{n} \ell_{i,j}(\ell_{i,j}/\bar{\ell}_i)^\lambda$, and selects the top $k$ models with the smallest $\ell_{h_j}$, where $k \leq m$. `MS` simplifies to `re-training` at $\lambda = 0$. The model with the smallest $\ell_{h_j}$ is used to compute the residuals for the next iteration, and we return the top $k$ models at the last boosting iteration. The intuition of this reweighting is that the loss contribution of sample $\mathbf{s}_i$ is rewarded at an exponential scaling factor $\lambda \geq 0$ when the model $h_j$ produces lower loss than the average for $\mathbf{s}_i$ over all models in $\mathcal{R}^m(\mathcal{H}, \mathcal{S}_t, h_t^*, \epsilon)$. In contrast, the `IE` method constructs $U$ ensembles $\overline{h_u}$, $u \in [U]$ in each iteration, where each ensemble consists of $E$ randomly selected

---

[12]MEO and SP respectively quantifies the discrepancy in the sum of True Positive Rate and False Positive Rate, and in the probability of the model outputting class 1.

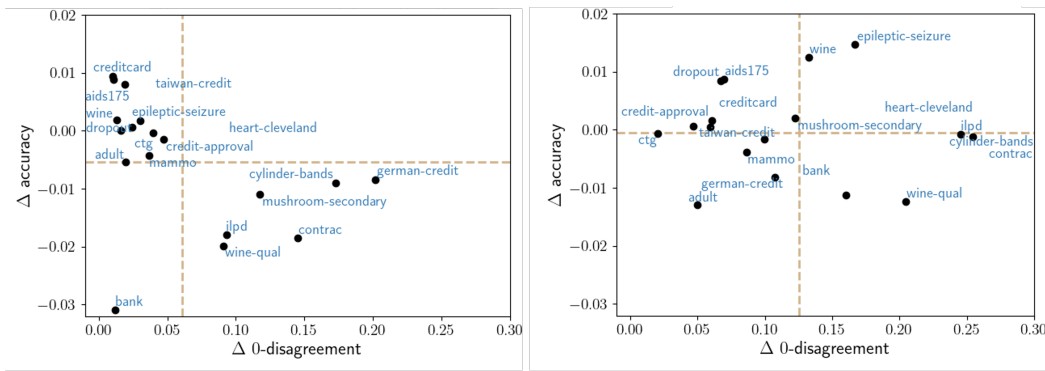

Figure 5: `re-training` vs. `MS` with $\lambda = 3$ (left) and `IE` with $E = 20$ (right) to mitigate predictive multiplicity on 18 UCI datasets. Each point is averaged over 20 random train-test splits (std. omitted for clarity). Dashed lines are the mean of each axis. *Higher values are better* for both axes.

models from $\mathcal{R}^m(\mathcal{H}, \mathcal{S}_t, h_t^*, \epsilon)$. The model $\overline{h_u}$ is constructed by an additive weighted sum of the outputs of the $E$ models, i.e., $\overline{h_u}(\mathbf{x}_i) = \frac{1}{E} \sum_{e=1}^{E} w_e h_e(\mathbf{x}_i)$, where the weights $w_e = (1/\ell_{h_e})/w$, and $w$ is the harmonic mean of all losses $\{\ell_{h_e}\}_{e=1}^{E}$.

For evaluation, we extend the sample-wise metric disagreement $\mu(\mathbf{s}_i)$ in Kulynych et al. [48] (cf. Appendix B.2) to consider the disagreement across all samples in the whole dataset. We call this new metric *p-disagreement*[13], defined as $d(\mathcal{S}, p) \triangleq \frac{1}{n} \sum_{i=1}^{n} \mathbb{1}\left[\mu(\mathbf{s}_i) \leq p\right]$. In Figure 5 we report the reduction of predictive multiplicity ($\Delta$0-disagreement) and improvement of accuracy ($\Delta$accuracy) vs. a `re-training` baseline, on 18 UCI tabular datasets [5]. In our experiments, $U = m = 100$, $k = 25$, and $E = 20$. For a fair comparison, `MS` and `IE` share the same training procedure as `re-training`—in all iterations we select the top $k = 25$ from $m = 100$ models.

We observe that `IE` outperforms `MS` in disagreement reduction while both methods yield a similar accuracy to `re-training`. However, `IE` has the cost of increasing the overall model complexity by a factor of $E = 20$, which may be undesirable for interpretability or auditing. It is noteworthy that a small $\Delta$accuracy could lead to a great reduction in disagreement. For example, in "epileptic-seizure", `re-training` has a 0-disagreement of $0.208$ and `IE` reduces it to $0.041$ with only a slight improvement of $0.014 \pm 0.002$ in accuracy. For ablation studies on the hyperparameters $E$, $k$ and $\lambda$ and more explanations, see Appendix E.10.

## 5 Discussion

Here we reflect on the limitations and highlight interesting avenues for future work.

**Limitations.** While `re-training` with different random seeds offers a "global" exploration of models within the Rashomon set, `RashomonGB` conducts a "local" exploration. Therefore, the effectiveness of `RashomonGB` depends on selecting diverse models in each iteration; similar models reduce its efficiency in exploring Rashomon sets. However, `RashomonGB` demonstrates greater efficiency than the `re-training` strategy, highlighting a trade-off between the efficiency of exploring the Rashomon set and the effectiveness of capturing model diversity. Finding the (sub-)optimal strategy for exploring the Rashomon set remains an active area of research. Our theoretical analysis, developed for gradient boosting, needs an extension to other algorithms like adaptive boosting to validate its applicability. Additionally, the large number of models generated by `RashomonGB` poses storage challenges, requiring new data structures for efficient management.

**Future directions.** First, our analysis in Section 3 links the size of the Rashomon set to dataset quality, providing a foundation for studying how dataset properties impact the Rashomon effect. Second, combining models in each `RashomonGB` iteration is similar to model stitching [50], suggesting potential insights if adapted for neural networks. Third, varying the number of models selected per iteration could enhance `RashomonGB`'s flexibility and effectiveness.

---

[13]$d(\mathcal{S}, 0)$ counts the fraction of samples that have zero disagreement among the models in the Rashomon set.

**Disclaimer.** This paper was prepared for informational purposes by the Global Technology Applied Research center and Artificial Intelligence Research group of JPMorgan Chase & Co. This paper is not a product of the Research Department of JPMorgan Chase & Co. or its affiliates. Neither JPMorgan Chase & Co. nor any of its affiliates makes any explicit or implied representation or warranty and none of them accept any liability in connection with this paper, including, without limitation, with respect to the completeness, accuracy, or reliability of the information contained herein and the potential legal, compliance, tax, or accounting effects thereof. This document is not intended as investment research or investment advice, or as a recommendation, offer, or solicitation for the purchase or sale of any security, financial instrument, financial product or service, or to be used in any way for evaluating the merits of participating in any transaction.

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

The appendix is divided into the following parts. Appendix A: Omitted proofs and theoretical results; Appendix B: Discussion on predictive multiplicity metrics; Appendix C: discussion on the Rashomon set with Rademacher complexity; Appendix D: Details on the experimental setup; and Appendix E: Additional empirical results and ablation studies.

## A Omitted proofs and theoretical results

We first introduce the following useful lemmas to facilitate the proofs of the propositions. The first lemma is a fundamental property of sub-Gaussian random variables (Proposition 2.5.2 from Vershynin [72]), and the second lemma is a slight extension of the Theorem 3 in Xu and Raginsky [77].

**Lemma A.1.** *If a random variable $X$ is $\sigma$-sub-Gaussian, then*

$$\Pr\left(|X| \geq \gamma\right) \leq 2e^{-\frac{\gamma^2}{\sigma^2}}. \tag{A.1}$$

**Lemma A.2.** *Suppose $\ell(h, S)$ is $\sigma$-sub-Gaussian under the data-generating distribution $P_S$ with $|\mathcal{S}| = n$ for all $h \in \mathcal{H}$, then*

$$\Pr\left(L_{P_S}(H) - L_S(H) > \alpha\right) \leq \beta, \tag{A.2}$$

*if $n = \frac{8\sigma^2}{\alpha^2}\left(\frac{I(S;H)}{\beta} + \log\frac{2}{\beta}\right)$ for any $\alpha > 0$ and $0 < \beta \leq 1$.*

### A.1 Proof of Proposition 1

By the definition of the Rashomon set in Eq. (1), we know that $h \in \mathcal{R}(\mathcal{H}, \mathcal{S}, h^*, \epsilon)$ if and only if the loss deviation is upper bounded by $\epsilon$; that is,

$$L_{P_S}(h) - L_{P_S}(h^*) \leq \epsilon. \tag{A.3}$$

We can decompose the loss deviation in Eq. (A.3) by

$$L_{P_S}(h) - L_{P_S}(h^*) = \underbrace{L_{P_S}(h) - L_S(h)}_{(1)} + \underbrace{L_S(h) - L_{P_S}(h^*)}_{(2)}. \tag{A.4}$$

By picking $\beta = \frac{\rho}{2}$ in Lemma A.2, we have with probability at least $1 - \frac{\rho}{2}$,

$$(1) = L_{P_S}(h) - L_S(h) \leq \sqrt{\frac{8\sigma^2}{n}\left(\frac{2I(S;H)}{\rho} + \ln\frac{4}{\rho}\right)}. \tag{A.5}$$

Moreover, since the loss function $\ell(h, S)$ is $\sigma$-sub-Gaussian, the risk $L_S(h)$ is $\sigma/\sqrt{n}$-sub-Gaussian. By Lemma A.1 and the non-negativity of the loss functions, we have

$$\Pr\left(L_S(h) - L_{P_S}(h^*) \geq \gamma\right) \leq \Pr\left(L_S(h) \geq \gamma\right) \leq 2e^{-\frac{n\gamma^2}{\sigma^2}}. \tag{A.6}$$

Let $2e^{-\frac{n\gamma^2}{\sigma^2}} = \frac{\rho}{2}$ and solve for $\gamma$, we have with probability at least $1 - \frac{\rho}{2}$,

$$(2) = L_S(h) - L_{P_S}(h^*) \leq \sqrt{\frac{\sigma^2}{n}\ln\frac{4}{\rho}}. \tag{A.7}$$

Therefore, combining Eq. (A.4), Eq. (A.5) and Eq. (A.7) with probability union bounds, we have with probability at least $1 - \rho$,

$$L_{P_S}(h) - L_{P_S}(h^*) \leq \sqrt{\frac{8\sigma^2}{n}\left(\frac{2I(S;H)}{\rho} + \ln\frac{4}{\rho}\right)} + \sqrt{\frac{\sigma^2}{n}\ln\frac{4}{\rho}}, \tag{A.8}$$

and hence

$$\Pr\left(h \in \mathcal{R}\left(\mathcal{H}, \mathcal{S}, h^*, \sqrt{\frac{8\sigma^2}{n}\left(\frac{2I(S;H)}{\rho} + \ln\frac{4}{\rho}\right)} + \sqrt{\frac{\sigma^2}{n}\ln\frac{4}{\rho}}\right)\right) > 1 - \rho. \tag{A.9}$$

## A.2 Proof of Proposition 2

We start with the Rashomon set of fitting models to the pseudo-residuals at $T$-th boosting iteration. With the MSE loss, we have $r_{ti} = -\left[\frac{\partial \ell(f_{t-1}, \mathbf{s}_i)}{\partial f_{t-1}}\right] = 2(y_i - f_{t-1}(\mathbf{x}_i))$. We neglect the factor 2 as it is just a constant. From Proposition 1, we know that for $h_T \in \mathcal{H}$, with probability at least $1 - \rho$,

$$\frac{1}{n}\sum_{i=1}^{n}(h_T(\mathbf{x}_i) - r_{ti})^2 \leq \frac{1}{n}\sum_{i=1}^{n}(h_T^*(\mathbf{x}_i) - r_{Ti})^2 + \epsilon_T, \tag{A.10}$$

where $\epsilon_T = \sqrt{\frac{8\sigma^2}{n}\left(\frac{2I(S_T;H)}{\rho} + \ln\frac{4}{\rho}\right)} + \sqrt{\frac{\sigma^2}{n}\ln\frac{4}{\rho}}$ is the Rashomon parameter. By plugging in the recursive relation of the pseudo-residuals, we have

$$\frac{1}{n}\sum_{i=1}^{n}(h_T^*(\mathbf{x}_i) - r_{Ti})^2 + \epsilon_T = \frac{1}{n}\sum_{i=1}^{n}(h_T^*(\mathbf{x}_i) - (y_i - \sum_{t=0}^{T-1}h_t^*(\mathbf{x}_i))))^2 + \epsilon_T$$

$$= \frac{1}{n}\sum_{i=1}^{n}(\sum_{t=0}^{T}h_t^*(\mathbf{x}_i) - y_i)^2 + \epsilon_t \tag{A.11}$$

$$= \frac{1}{n}\sum_{i=1}^{n}(f_T^*(\mathbf{x}_i) - y_i)^2 + \epsilon_t.$$

On the other hand, for the left-handed term in Eq. (A.10), we have for $h_{T-1} \in \mathcal{H}$, with probability at least $1 - \rho$

$$\frac{1}{n}\sum_{i=1}^{n}(h_T(\mathbf{x}_i) - r_{ti})^2 = \frac{1}{n}\sum_{i=1}^{n}(h_T(\mathbf{x}_i) - r_{(T-1)i} + h_{T-1}^*(\mathbf{x}_i))^2$$

$$\geq \frac{1}{n}\sum_{i=1}^{n}(h_T(\mathbf{x}_i) - r_{(T-1)i} + h_{T-1}(\mathbf{x}_i))^2 - \epsilon_{T-1} \tag{A.12}$$

$$= \frac{1}{n}\sum_{i=1}^{n}(\sum_{t=T-1}^{T}h_t(\mathbf{x}_i) - r_{(T-1)i})^2 - \epsilon_{T-1},$$

where the inequality in Eq. (A.12) comes again from Proposition 1. By repeating the decomposition of the pseudo-residuals in Eq. (A.12), we have for $f_T \in \mathcal{F}$, with probability at least $1 - (T-1)\rho$,

$$\frac{1}{n}\sum_{i=1}^{n}(h_T(\mathbf{x}_i) - r_{ti})^2 \geq \frac{1}{n}\sum_{i=1}^{n}(\sum_{t=1}^{T}h_t(\mathbf{x}_i) - y_i)^2 - \sum_{t=1}^{T-1}\epsilon_t$$

$$= \frac{1}{n}\sum_{i=1}^{n}(f_T(\mathbf{x}_i) - y_i)^2 - \sum_{t=1}^{T-1}\epsilon_t. \tag{A.13}$$

Finally, combining Eq. (A.10), Eq. (A.11), and Eq. (A.13) with probability union bounds, we have for $f_T \in \mathcal{F}$, with probability at least $1 - T\rho$

$$\frac{1}{n}\sum_{i=1}^{n}(f_T(\mathbf{x}_i) - y_i)^2 \leq \frac{1}{n}\sum_{i=1}^{n}(f_T^*(\mathbf{x}_i) - y_i)^2 + \sum_{t=1}^{T}\epsilon_t, \tag{A.14}$$

where the overall Rashomon parameter $\sum_{t=1}^{T}\epsilon_t = T\sqrt{\frac{\sigma^2}{n}\ln\frac{4}{\rho}} + \sum_{t=1}^{T}\sqrt{\frac{8\sigma^2}{n}\left(\frac{2I(S_t;H)}{\rho} + \ln\frac{4}{\rho}\right)}$.

## A.3 Proof of Proposition 3

With loss of generality, let $t_1 = t - 1$ and $t_2 = t$. If the loss function $\ell$ is the MSE loss, the pseudo-residuals of the $t$-th boosting iteration have the form

$$r_{ti} = -\left[\frac{\partial \ell(f_{t-1}, \mathbf{s}_i)}{\partial f_{t-1}}\right] = 2(y_i - f_{t-1}(\mathbf{x}_i)) = 2(y_i - f_{t-2}(\mathbf{x}_i) - h_{t-1}(\mathbf{x}_i))$$

$$= r_{(t-1)i} - 2h_{t-1}(\mathbf{x}_i). \tag{A.15}$$

Therefore, we have the recursive relation of the residual variables as $R_t = R_{t-1} - 2H$. The mutual information then follows as

$$I(H; R_t|X) = I(H; R_{t-1}, H|X) = I(H; R_{t-1}|X) + I(H; H|X, R_{t-1}) \geq I(H; R_{t-1}|X),$$
(A.16)

since mutual information $I(H; H|X, R_{t-1})$ is non-negative.

Similarly, if the loss function is the binary CE loss, the pseudo-residuals of the $t$-th boosting iteration have the form

$$r_{ti} = -\left[\frac{\partial \ell(f_{t-1}, \mathbf{s}_i)}{\partial f_{t-1}}\right] = \frac{y_i}{f_{t-1}(\mathbf{x}_i)} - \frac{1 - y_i}{1 - f_{t-1}(\mathbf{x}_i)} = \frac{y_i - f_{t-1}(\mathbf{x}_i)}{f_{t-1}(\mathbf{x}_i)(1 - f_{t-1}(\mathbf{x}_i))}$$
$$= \frac{y_i - f_{t-2}(\mathbf{x}_i)}{f_{t-2}(\mathbf{x}_i)(1 - f_{t-2}(\mathbf{x}_i))} + c = r_{(t-1)i} + c$$
(A.17)

where $c$ is a function of $h_{t-1}(\mathbf{x}_i)$:

$$\frac{h_{t-1}(\mathbf{x}_i)f_{t-2}(\mathbf{x}_i)(1 - f_{t-2}(\mathbf{x}_i))(1 - 2f_{t-2}(\mathbf{x}_i) - h_{t-1}(\mathbf{x}_i))(f_{t-2}(\mathbf{x}_i) - y_i) - h_{t-1}(\mathbf{x}_i)}{f_{t-2}(\mathbf{x}_i)(1 - f_{t-2}(\mathbf{x}_i)) + h_{t-1}(\mathbf{x}_i)(1 - 2f_{t-2}(\mathbf{x}_i) - h_{t-1}(\mathbf{x}_i))}.$$
(A.18)

Therefore, consider the corresponding random variables of Eq. (A.17), we have $R_t = R_{t-1} + C$ and the desired result follows from Eq. (A.16).

# B  Additional discussions

We provide addition background and discussions on gradient boosting, predictive multiplicity, the Rashomon gradient boosting algorithm, and group fairness.

## B.1  More on gradient boosting

The minimization of $L_{\mathcal{S}}(f_T)$ can be viewed as functional gradient descent (FGD), which is a computationally infeasible optimization problem in general [58]. A common alternative to approximate a FGD is applying steepest descent to find a local minimum of the loss function by iterating on a given $f_{t-1}(\mathbf{x})$, i.e., $f_t(\mathbf{x}) = f_{t-1}(\mathbf{x}) - \eta \nabla_{f_{t-1}} L_{\mathcal{S}}(f_{t-1})$ with a learning rate $\eta \in \mathbb{R}^+$. This way, in a forward stage-wise manner, we can fit a basis function $h_t \in \mathcal{H}$ that is closest to the functional gradient $\nabla_{f_{t-1}} L_{\mathcal{S}}(f_{t-1})$ subject to a distance measure $d$ with $h_t = \mathrm{argmin}_{h \in \mathcal{H}} d(\nabla L_{\mathcal{S}}(f_{t-1}), h)$. Varying the choices of $\ell$ and $d$ recovers most widely-used boosting algorithms. For instance, if $\ell(h_t, \mathbf{s}_i) = e^{-y_i h_t(\mathbf{x}_i)}$ and $d(\nabla L_{\mathcal{S}}(f_{t-1}), h_t) = -\nabla L_{\mathcal{S}}(f_{t-1}) \cdot h_t$, the FGD recovers adaptive boosting (AdaBoost) [32], and if $\ell(h_t, \mathbf{s}_i) = \log(1 + e^{-y_i h(\mathbf{x}_i)})$, it recovers LogitBoost [33]. Finally, if $d$ is the $\ell_2$-norm $\| - \nabla L_{\mathcal{S}}(f_{t-1}) - h_t \|_2^2$ instead of the inner product, it recovers the gradient boosting [34]. One of the most popular variants of gradient boosting is by further applying LASSO and ridge regularizations, called the eXtreme Gradient Boosting (XGB) [15].

## B.2  More on predictive multiplicity and the empirical Rashomon sets

We summarize existing predictive multiplicity metrics, from their background, mathematical formulation, operational meanings, to computational details. Predictive multiplicity metrics can be categorized into two groups: score-based and decision-based, where a decision is a thresholded score or the score vector after $\mathrm{argmax}$. Precisely, consider a binary classification, if we have a score $q$, then the decision can be obtained by $\mathbb{1}[s > \tau]$, where $\tau$ is a threshold. For a $c$-class classification problem where $c > 2$, the score is a vector, say $\mathbf{q} \in \Delta_c$, and the decision can be obtained by $\mathrm{argmax}_{i \in [c]} [\mathbf{q}]_i$. In the following, we start with the decision-based metrics, see Table B.1.

Decision-based predictive multiplicity metrics essentially measure the "conflictions" of the decisions either for the whole dataset or per sample. Marx et al. [56] propose two metrics: ambiguity and discrepancy; both of them measure the fraction of conflicting decisions across a dataset. Ambiguity is the proportion of samples in a dataset that can be assigned conflicting predictions by competing classifiers in the Rashomon set. Discrepancy is the maximum number of predictions that could change in a dataset if we were to switch between models within the Rashomon set. More precisely, given a pre-trained model $h_{\mathbf{w}^*}$, the ambiguity $\alpha(\mathcal{D})$ and the discrepancy $\delta(\mathcal{D})$ are respectively defined in [56, Definitions 3 and 4]. Both ambiguity and discrepancy can be estimated by a mixed integer program [56, Section 3]. The implementation of estimating ambiguity and discrepancy can be accessed at https://github.com/charliemarx/pmtools.

Table B.1: Decision-based predictive multiplicity metrics.

| Metrics | Definitions |
|---|---|
| Ambiguity [56] | $\alpha(\mathcal{D}) \triangleq \frac{1}{|\mathcal{D}|} \sum_{\mathbf{x}_i \in \mathcal{D}} \max_{h_{\mathbf{w}} \in \mathcal{R}} \mathbb{1}\left[\mathrm{argmax}\, h_{\mathbf{w}}(\mathbf{x}_i) \neq \mathrm{argmax}\, h_{\mathbf{w}^*}(\mathbf{x}_i)\right]$ |
| Discrepancy [56] | $\delta(\mathcal{D}) \triangleq \max_{h_{\mathbf{w}} \in \mathcal{R}} \frac{1}{|\mathcal{D}|} \sum_{\mathbf{x}_i \in \mathcal{D}} \mathbb{1}\left[\mathrm{argmax}\, h_{\mathbf{w}}(\mathbf{x}_i) \neq \mathrm{argmax}\, h_{\mathbf{w}^*}(\mathbf{x}_i)\right]$ |
| Disagreement [10, 48] | $\mu(\mathbf{x}_i) \triangleq 2\Pr\{\mathrm{argmax}\, h_{\mathbf{w}}(\mathbf{x}_i) \neq \mathrm{argmax}\, h'_{\mathbf{w}}(\mathbf{x}_i); h_{\mathbf{w}}, h'_{\mathbf{w}} \in \mathcal{R}\}$ |

Table B.2: Score-based predictive multiplicity metrics.

| Metrics | Definitions |
|---|---|
| Std./ Var. of scores [52, 17, 9] | $s(\mathbf{x}_i) \triangleq \sqrt{\mathbb{E}_{h_{\mathbf{w}} \sim P_{\mathcal{R}}}\left[(h_{\mathbf{w}}(\mathbf{x}_i) - \mathbb{E}_{h_{\mathbf{w}} \sim P_{\mathcal{R}}}[h_{\mathbf{w}}(\mathbf{x}_i)])^2\right]}$ |
| Viable Prediction Range (VPR) [74] | $v(\mathbf{x}_i) \triangleq \max_{h_{\mathbf{w}} \in \mathcal{R}} h_{\mathbf{w}}(\mathbf{x}_i) - \min_{h_{\mathbf{w}} \in \mathcal{R}} h_{\mathbf{w}}(\mathbf{x}_i)$ |
| Rashomon Capacity (RC) [42] | $c(\mathbf{x}_i) \triangleq \sup_{P_{\mathcal{R}}} \inf_{\mathbf{q} \in \Delta_c} \mathbb{E}_{h_{\mathbf{w}} \sim P_{\mathcal{R}}} D_{\mathsf{KL}}(h_{\mathbf{w}}(\mathbf{x}_i) \| \mathbf{q})$ |

Instead of computing the empirical fraction of conflicting decision over a dataset $\mathcal{D}$, disagreement directly using the probability of the occurrence of conflicting decisions per sample Black et al. [10, Section A.1] and Kulynych et al. [48, Eq. (4)]. The factor 2 in the definition of disagreement ensures that $\mu(\mathbf{x}_i)$ is in the $[0, 1]$ range for the ease of interpretation. Kulynych et al. [48] further proposed a plug-in estimator to estimate disagreement for binary classification with a sample complexity bound on the number of models obtained by re-training. The implementation of directly estimating disagreement from the empirical Rashomon set along with the plug-in estimator can be accessed at https://github.com/spring-epfl/dp_multiplicity

On the other hand, score-based metrics focus on the spread of the output scores; see Table B.2. The most straightforward metric is to compute the standard deviation (std.) $s(\mathbf{x}_i)$ (and the variance (var.)) of the scores of a sample by all models in the Rashomon set Long et al. [52, Definition 2], see https://github.com/Carol-Long/Fairness_and_Arbitrariness for the implementation. However, score std. or var. fails to capture large score spreads that concentrate on a small subset of models. To precisely capture the largest possible spread of scores, Watson-Daniels et al. [74, Definition 2] proposed Viable Prediction Range (VPR) $v(\mathbf{x}_i)$, which is the largest score deviation of a sample that can be achieved by models in the Rashomon set. The VPR can be computed using similar mixed integer programs in Marx et al. [56] for binary classification with linear classifiers. However, Watson-Daniels et al. [74] did not release their codes.

Borrowing from information theory, Hsu and Calmon [42, Definition 2] measures the spread of output scores for $c$-class classification problems in the probability simplex $\Delta_c$ by an analog of channel capacity, termed the Rashomon Capacity. Note that the infimum $\inf_{\mathbf{q} \in \Delta_c} \mathbb{E}_{h_{\mathbf{w}} \sim P_{\mathcal{R}}} D_{\mathsf{KL}}(h_{\mathbf{w}}(\mathbf{x}_i) \| \mathbf{q})$ measures (in the sense of KL divergence) the spread of the scores of a sample $\mathbf{x}_i$ given a distribution $P_{\mathcal{R}}$ over all the models $h_{\mathbf{w}}$ in the Rashomon set, where the minimizing $q$ acts as a "centroid" for the outputs of the classifiers. The supremum picks the worst-case distribution $P_{\mathcal{R}}$ over all possible distributions in the Rashomon set. They proposed the adversarial weight perturbation (AWP), which perturbs the weights of a pre-trained model such that the output scores of a sample are thrust toward all possible classes. The outputs of the perturbed models can then be used to compute RC by the Blahut-Aromoto algorithm [11, 4]. The implementation of AWP and the Blahut-Aromoto algorithm can be accessed at https://github.com/HsiangHsu/rashomon-capacity.

The estimation of all these predictive multiplicity metrics is, in practice, computed with the empirical Rashomon set, as the true Rashomon set in Eq. (1) is computationally infeasible. Note that, however, the size of the empirical Rashomon set is not a proxy for multiplicity. For instance, an empirical Rashomon set with 100 globally diverse (e.g., obtained by re-training with different seeds) models might exhibit a higher predictive multiplicity metric (e.g., VPR) compared to another empirical Rashomon set containing 1000 models that differ only locally. The size of the true Rashomon set, on the other hand, representing an ideal scenario achievable with unlimited computational and storage resources, can indeed act as a proxy for predictive multiplicity. In this context, predictive multiplicity metrics are non-decreasing with a larger size of the true Rashomon set (i.e., a larger $\epsilon$).

### B.3  More on `RashomonGB`

We provide visualizations of `RashomonGB` in Figure B.6. For the sake of illustration, we pick $T = 3$ and $m = 2$; however, $T$ and $m$ could be arbitrary numbers. The left hand side shows $2$ re-training of gradient boosting, where the final models are $M_{31}$ and $M_{32}$, and there are $m \times T = 2 \times 3 = 6$

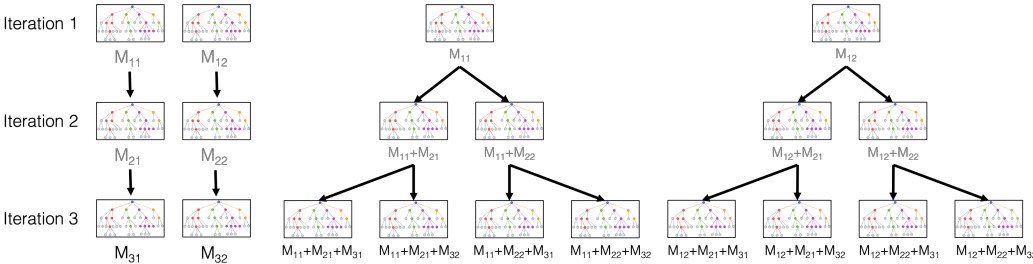

Figure B.6: `Re-training` and `RashomonGB` with the same training costs produce difference numbers of models in the Rashomon set.

training in total. The right hand side is `RashomonGB` that generates $m^T = 2^3 = 8$ models, i.e., $\{M_{11}, M_{12}\} + \{M_{21}, M_{22}\} + \{M_{31}, M_{32}\}$. In essence, building the Rashomon set with `RashomonGB` is similar to expanding the tree in each iteration (i.e., layers) until it is a complete $m$-ary tree.

## B.4 Group-fairness metrics and fairness interventions

Consider a triple of sample $(\mathbf{x}, a, y)$, where $\mathbf{x}$ and $y$ are the usual feature and label pair, and $a$ is an additional attribute, such as race or sex, that can be used to split the datasets into different groups (e.g., female vs. male if $a$ is sex). Group fairness is measured by the gap of a certain quantity evaluated on different groups. For example, statistical parity (SP) considers the the probability of the model outputting class 1, and mean equalized odds (MEO) considers the sum of True Positive Rate (TPR) and False Positive Rate (FPR). For simplicity, denote $\hat{Y} \operatorname{argmax} h(X)$ the prediction label from the model $h$, and assume that there are $B$ groups. Then the equalized odds (EO) and (SP) can be formally defined as:

**Definition B.1** (SP [29]). $\Pr\left(\hat{Y} = 1 | A = a\right) = \Pr\left(\hat{Y} = 1 | A = a'\right), \forall a, a' \in [B]$.

**Definition B.2** (EO [38]). $\Pr\left(\hat{Y} = 1 | A = a, Y = y\right) = \Pr\left(\hat{Y} = 1 | A = a', Y = y\right), \forall a, a' \in [B]$ *and* $\forall y \in [c]$.

The operational meaning of the two group fairness metrics is as follows: Statistical Parity (SP) requires that the predicted label $\hat{Y}$ be independent of the group attribute $A$, while Equalized Odds (EO) conditions on both the group attribute and the true label for independence. EO improves upon SP by allowing for perfect classifiers when the true label $Y$ is correlated with the group attribute $A$. In practice, we measure fairness by quantifying the level of EO and SP violations, as reported in Figure 4. The violation of EO is also referred to as the mean EO.

$$
\begin{aligned}
\text{MEO} &\triangleq \max_{a,a' \in [B]} \frac{1}{2} \left( |\text{TPR}_{A=a} - \text{TPR}_{A=a'}| + |\text{FPR}_{A=a} - \text{FPR}_{A=a'}| \right). \\
\text{SP violation} &\triangleq \max_{a,a' \in [B]} \frac{1}{2} \left| \Pr\left(\hat{Y} = 1 | A = a\right) - \Pr\left(\hat{Y} = 1 | A = a'\right) \right|.
\end{aligned} \tag{B.19}
$$

Fairness intervention algorithms aim to make the outputs of a machine learning model satisfy either MEO or SP violation smaller than a given budget. The fairness interventions can be categorized into three categories: pre-processing, in-processing and post-processing. Pre-processing mechanisms, such as the one proposed by Calmon et al. [14], transform the dataset using a random mapping to reduce group fairness metrics while preserving utility. This approach is the most flexible within the data science pipeline, as it is independent of the modeling algorithm and can be integrated with data release and publishing mechanisms. In-processing mechanisms such as `Reduction` incorporate fairness constraints directly into the training process. This typically involves adding a fairness constraint to the loss function, resulting in a fair classifier. Post-processing mechanisms such as `EqOdds` and `rejection` treat the model as a black box and adjust its predictions (by, e.g., tilting) to meet the desired fairness constraints.

Here, we provide more details on the fairness intervention baselines used in Section 4.2. `Reduction` [1], short for exponentiated gradient reduction, is an in-processing technique that converts fair classification into a sequence of cost-sensitive classification problems. It produces a randomized classifier that achieves the lowest empirical error while satisfying the desired fairness constraints. Although this technique effectively achieves fairness with minimal accuracy loss, it is computationally expensive due to the need for re-training multiple models. `Rejection` [45] is a post-processing technique that achieves fairness constraints by adjusting the outcomes of samples within a confidence band around the decision boundary. It assigns favorable outcomes to unprivileged groups and unfavorable outcomes to privileged groups, resulting in thresholded predictions rather than probabilities over binary labels. `EqOdds` [38] is a post-processing technique that addresses empirical risk minimization with a fairness constraint by formulating it as a linear program. It adjusts predictions based on the derived probabilities to achieve equalized odds. `FaiRS` [20] develops a framework for characterizing predictive fairness properties across models in the Rashomon set. They also propose a variant of `FaiRS` that addresses the issue of selective labels, achieving the same guarantees with oracle access to the outcome regression function.

# C  Characterizing Rashomon sets with Rademacher complexity

Besides the information-theoretic analysis on the Rashomon sets in Section 3, here, we further provide an analysis on the size the Rashomon sets based on Rademacher complexity [68]. Similar to the Vapnik–Chervonenkis (VC) dimension, Rademacher complexity measures the richness of a set of functions with respect to a probability distribution, reflecting its capacity to fit random noise, and is widely-adopted in modern analysis of machine learning generalizability. Following the notations in Section 2, the definition of Rademacher complexity is as follows.

**Definition C.3** (Rademacher Complexity). *For a hypothesis space $\mathcal{H}$, samples $\mathcal{S}$ and a loss function $\ell$, the Rademacher complexity of $\mathcal{H}$ with respect to $\mathcal{S}$ is defined as*

$$\mathsf{Rad}(\ell \circ \mathcal{H} \circ \mathcal{S}) \triangleq \frac{1}{n} \mathbb{E}_{\sigma \sim \{\pm 1\}^n} \left[ \sup_{h \in \mathcal{H}} \sum_{i=1}^{n} \sigma_i \ell(h, \mathbf{s}_i) \right], \tag{C.20}$$

*where the random variable in $\sigma$ are i.i.d. distributed according to the Rademacher distribution, i.e., $P(\sigma_i = 1) = P(\sigma_i = -1) = 0.5$.*

The following lemma is one of the fundamental results of the Rademacher Complexity.

**Lemma C.3** ([68, Lemma 26.5]). *Assume for all $\mathbf{s}_i \in \mathcal{S}$ and $h \in \mathcal{H}$ we have $|\ell(h, \mathbf{s}_i)| \leq M$, i.e., the loss function is bounded by $M$; then with probability of at least $1 - \rho$, for all $h \in \mathcal{H}$,*

$$L_{P_S}(h) - L_{\mathcal{S}}(h) \leq 2\mathsf{Rad}(\ell \circ \mathcal{H} \circ \mathcal{S}) + 4M\sqrt{\frac{2\ln(4/\rho)}{n}}. \tag{C.21}$$

The following proposition states that the size of a Rashomon set is controlled by the Rademacher complexity.

**Proposition C.4.** *Let $\gamma \triangleq \sup_{h, h' \in \mathcal{H}} L_{\mathcal{S}}(h) - L_{\mathcal{S}}(h')$, we have with probability of at least $1 - \rho$,*

$$h \in \mathcal{R}\left(\mathcal{H}, \mathcal{S}, h^*, \gamma + 2\mathsf{Rad}(\ell \circ \mathcal{H} \circ \mathcal{S}) + 5M\sqrt{\frac{2\ln(4/\rho)}{n}}\right). \tag{C.22}$$

*Proof.* Recall the constrain in the definition of the Rashomon set in Eq. (1), i.e.,

$$L_{P_S}(h) - L_{P_S}(h^*) = \underbrace{L_{P_S}(h) - L_{\mathcal{S}}(h)}_{(1)} + \underbrace{L_{\mathcal{S}}(h) - L_{\mathcal{S}}(h^*)}_{(2)} + \underbrace{L_{\mathcal{S}}(h^*) - L_{P_S}(h^*)}_{(3)}. \tag{C.23}$$

Term (1) follows directly from Lemma C.3 and let $(2) \leq \sup_{h \in \mathcal{H}} L_{\mathcal{S}}(h) - L_{\mathcal{S}}(h^*) \leq \gamma$ by assumption. By the Hoeffding's inequality [41], and denote $z_i = \ell(h, \mathbf{s}_i)$, we have for any function $h \in \mathcal{H}$,

$$P\left(|L_{\mathcal{S}}(h) - L_{P_S}(h)| \leq t\right) = P\left(\left|\frac{1}{n}\sum_{i=1}^{n} z_i - \mathbb{E}\left[\frac{1}{n}\sum_{i=1}^{n} z_i\right]\right| \leq t\right) \leq 2\exp\left(-\frac{2nt^2}{M^2}\right). \tag{C.24}$$

Therefore by taking $h = h^*$, with probability of at least $1 - \rho/2$, we have

$$(3) = L_{\mathcal{S}}(h^*) - L_{P_S}(h^*) \leq M\sqrt{\frac{\ln(4/\rho)}{2n}}. \tag{C.25}$$

Combining all together with the union bound, we have with probability of at least $1 - \rho$

$$L_{P_S}(h) - L_{P_S}(h^*) \leq \gamma + 2\mathsf{Rad}(\ell \circ \mathcal{H} \circ \mathcal{S}) + 5M\sqrt{\frac{2\ln(4/\rho)}{n}}. \tag{C.26}$$

$\square$

If we pick the loss function to be the $p$-norm, i.e., $\ell(h(\mathbf{x}_i), y_i) = |h(\mathbf{x}_i) - y_i|^p$, then the results in Proposition C.4 become

$$P\left(h \in \mathcal{R}\left(\mathcal{H}, \mathcal{S}, h^*, \gamma + 2pM^{p-1}\mathsf{Rad}(\ell \circ \mathcal{H} \circ \mathcal{S}) + 5M^p\sqrt{\frac{2\ln(4/\rho)}{n}}\right)\right) \geq 1 - \rho. \tag{C.27}$$

Despite that we did not use Proposition C.4 in the main text, Eq. (C.22) stills provide us insights on controlling the size of Rashomon sets. For example, $\gamma$ and $M$ reflect the boundedness of the loss function, implying that CE loss, which is in general unbounded, could be a cause of a large Rashomon set. Moreover, the Rademacher complexity $\mathrm{Rad}(\ell \circ \mathcal{H} \circ \mathcal{S})$ suggests that a larger hypothesis space leaves the models a bigger "wiggle room" for multiplicity. Finally, note that as $n \to \infty$, the Rashomon parameter in Eq. (C.22) does no converge to $0$.

# D Details on the experimental setup

We summarize the dataset descriptions and training setups in Section 4.

## D.1 Dataset description and pre-processing

In this paper, we use 18 tabular datasets from the UCI machine learning repository [5], the ACS Income dataset [24] and the COMPAS recidivism dataset [3]. We summarize the descriptions of all tabular datasets, including the number of features, training/test split (seed $= 42$), and the label description in Table D.3.

The UCI machine learning repository (accessible at DuBois [28]; license: CC BY 4.0) is a well-known and widely used collection of 650 datasets for machine learning research and experimentation, and contains a diverse and extensive collection of datasets across various domains. We select 18 datasets in specific domains, including medicine, economics, society, etc., that may possess critical consequences if predictive multiplicity is not accounted for. For these UCI datasets, we remove samples with missing values, one-hot encoded nominal features, re-scale numeric features, and set the target label name to be 1 and the rest to be 0.

The ACS Income dataset (accessible at https://github.com/fairlearn/fairlearn; license: MIT license) and the UCI Adult dataset are collected from the United States Census Bureau. The goal of both datasets is to predict the income with demographic features of people. The UCI Adult dataset has a smaller size and a binary record of the income label ($Y = 1$ if income $> \$50K$ and $Y = 0$ otherwise). The ACS Income dataset has more than $1.6$ million samples. The income attribute is real-valued rather than a binary label. We transform the income to a binary labels by $Y = 1$ if income $> \$39K$ (the median income) and $Y = 0$ otherwise.

The COMPAS Recidivism dataset (accessible at https://www.propublica.org/datastore/dataset/compas-recidivism-risk-score-data-and-analysis; license: CC-BY-4.0) contains the prior criminal history for criminal defendants and the demographic makeup of prisoners in Brower County, Florida from 2013-2014. we select gender, age, number of prior crimes, length of custody and likelihood of recidivism to be the features. We pre-process the dataset by dropping missing/incomplete records, and convert categorical variables by one-hot encoding. For the group fairness experiments in Section 4.2, we only keep two races, African American (S = 0) and Caucasian (S = 1).

CIFAR-10 dataset (accessible at https://www.cs.toronto.edu/~kriz/cifar.html; license: MIT License) contains 60,000 images and equally distributed to 10 classes, such as cars, cats, etc. The dataset is split into 50,000 and 10,000 images for training and validation, respectively. Each image is a colored image with size of $32 \times 32$. For the pre-processing, we normalize each channel of image by the mean and standard deviation of the whole training set.

## D.2 Training setups and results

The weak learners used for all tabular datasets are decision trees with the Python Scikit Learn package [62]. The decision trees are expanded up to a maximal depth of 2, and the criterion for splitting is the squared loss with a impurity gain greater than $10^{-7}$. The minimum number of samples required to split an internal node is 2 and the maximum number of leaf nodes is 3. The number of iterations for `re-training` and `RashomonGB` is $T = 10$. In each iteration we train models with different random seeds for the splitting at each depth, and use a filtering process in place that screens out models with an MSE loss greater than 0.1 (i.e., $\epsilon_t = 0.1$) and retains models with an MSE loss smaller than 0.01 until $m = 10$ models are collected at each iteration. For `RashomonGB`, we randomly pick 2 out of $m$ models in each iteration. For all experiments, we set the learning rate to be $\alpha = 0.8$, and the loss function is the binary cross-entropy loss $L^{\mathsf{CE}}(h) \triangleq \frac{1}{n} \sum_{i=1}^{n} [-y_i \log \mathsf{softmax}(h(\mathbf{x}_i)) - (1 - y_i) \log(1 - \mathsf{softmax}(h(\mathbf{x}_i)))]$. For experiments in Section 4.2, we use the AIF360 [7] and Fairlearn [8] packages to implement the fairness intervention algorithms.

The weaker learner used for CIFAR-10 experiment is a 3-stage convolutional neural network with Python Pytorch package [61], each stage is composed by a convolutional layer with kernel size $3 \times 3$ and output channel 64, a batch normalization layer and a ReLU layer, and the residual shortcut is

added before ReLU for stage 2 and 3; at the end of each stage, an average pooling layer reduces the spatial dimension by 2 with kernel size 3. After stage 3, a global average pooling is applied to reduce the spatial dimension to $1\times1$, following by a linear layer for regression the pseudo-residual of each class. For each weaker learner, we train with batch size 256 for 50 epochs, and we use the AdamW [54] optimizer with fixed learning rate, $10^{-3}$ and weight decay, $10^{-4}$. Due to its complexity, we use $T = 6$ and $m = 50$ for both `re-training` and `RashomonGB`. During inference for `RashomonGB`, we random pick 3 out of $m$ models for each boosting iteration but 4 for the last iterations, which results in 972 models. We adopt the same learning rate $(0.8)$ used in the experiments for the tabular datasets.

Table D.3: Tabular dataset descriptions.

| Dataset | # of features | Training set size | Test set size | Label (# of classes) |
|---|---|---|---|---|
| ACS Income | 10 | 1331k | 332k | income larger than median or not (2) |
| UCI Adult | 104 | 22621 | 7541 | income >50K (2) |
| AIDS-175 | 26 | 1283 | 856 | patient death within the study period (2) |
| Bank Marketing | 63 | 30891 | 10297 | has deposit (2) |
| Cardiotocography (ctg) | 84 | 1275 | 851 | normal or not (2) |
| COMPAS | 6 | 4222 | 1056 | commit a crime again or not (2) |
| Contraception | 9 | 1104 | 369 | long or short term (2) |
| Credit Approval | 51 | 414 | 276 | credit card application approval or not (2) |
| Credit Card | 23 | 24000 | 6000 | default a payment or not (2) |
| Cylinder bands | 39 | 324 | 216 | band or no band (2) |
| Dropout | 36 | 2654 | 1770 | student drops out of school or not (2) |
| Epileptic seizure | 178 | 6900 | 4600 | subject has seizure or not (2) |
| German credit | 20 | 600 | 400 | good/bad credit risk (2) |
| Heart Disease (Cleveland) | 13 | 181 | 122 | absence/presence of heart disease (2) |
| ILPD | 10 | 349 | 234 | patient with/without liver disease (2) |
| Mammography | 5 | 622 | 208 | benign or malignant (2) |
| Mushroom Secondary | 20 | 36641 | 24428 | poisonous or not (2) |
| Qualitative Bankruptcy | 18 | 150 | 100 | had bankruptcy or not (2) |
| Taiwan Credit | 23 | 18000 | 12000 | borrower defaults on payment or not (2) |
| Wine | 13 | 106 | 72 | Wine type 2 vs. rest (2) |
| Wine Quality | 13 | 3898 | 2599 | Quality > 5 (2) |

# E   Additional results and experiments

We include an illustration on how to interpret Figure 3, additional experiments on other UCI datasets (UCI Adult, Bank Marketing, Mammography), computational time comparison between `re-training` and `RashomonGB`, ablation studies on different types of weak learners, number of boosting iterations $T$, number of models in each iteration $m$, comparison with predictive uncertainty estimation methods, and hyperparameters for mitigation predictive multiplicity ($E$, $k$, and $\lambda$).

## E.1   How to interpret Figure 3

We provide an explanation on how to properly interpret Figure 3 in Figure E.7. We first perform re-training with different random seeds and perform RashomonGB by using the same weak learners, i.e., the training cost of RashomonGB and Re-training are the same and hence the comparison presented in the paper is fair. Note that the $\epsilon$ we report here is the overall Rashomon parameter after $T = 10$ iterations, i.e., $T \times \epsilon$. Moreover, we do not compare models with different $\epsilon$ as it is clearly unfair. For the experiments of reporting predictive multiplicity in Section 4.1, we report the Rashomon parameter $\epsilon$ in the vertical axis (leftmost column) in Figure 3. For the experiments of fair model selection, we report the $\epsilon$ in the caption of Figure 4. For the experiments of mitigating predictive multiplicity by model averaging, we report the $\epsilon$ (in terms of the improvement of accuracy) in the vertical axis in Figure 5.

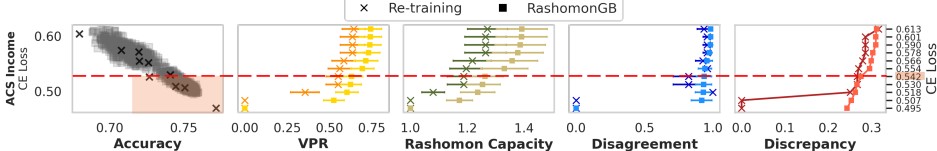

Figure E.7: This figure is part of the Figure 3. For illustration, we draw a red dashed line at CE Loss = 0.542. In the leftmost column, each marker represents a model for `Re-training` (cross) and `RashomonGB` (square), and the shaded area covers all models from either `Re-training` or `RashomonGB` that has CE Loss smaller than 0.542. In other words, cross and square markers in the shaded area form the Rashomon set building by `Re-training` and `RashomonGB` respectively. The values of predictive multiplicity metrics on the rightmost 4 columns at the intersection of the red dashed line are the predictive multiplicity metrics estimated with the Rashomon set of $L_{P_S}(f_T^*) + T\epsilon = 0.542$. Scanning the red dashed line upward leads to a larger Rashomon set. We would like to emphasize that our comparison between `Re-training` and `RashomonGB` is fair as those models are obtained from the same training cost.

## E.2 Estimating predictive multiplicity metrics of other UCI datasets

We include additional results in Figure E.8 on comparing the effectiveness of estimation prediction multiplicity metrics between `re-training` and `RashomonGB` (similar to Figure 3) for UCI Adult, Bank marketing, and Mammography datasets [5]. The UCI Adult dataset aims to predict whether the income of an individual exceeds 50,000 per year based on 1994 census data. The Bank Marketing dataset is related with direct marketing campaigns of a Portuguese banking institution based on phone calls in order to predict if the client will subscribe a bank term deposit or not. The Mammography dataset aims to discriminate between benign and malignant mammographic masses based on BI-RADS attributes and the patient's age. Our observation is that for decision-based predictive multiplicity metrics such as disagreement and discrepancy, `RashomonGB` outperforms `re-training` in terms of the effectiveness. For score-based predictive multiplicity metrics, when the CE loss constraint is small, `RashomonGB` performs better for all three datasets. When CE loss constraint is large, the size of the Rashomon set grows fast, and `re-training` captures more diverse models in the Rashomon set. These observations are consistent with the results shown in the main text.

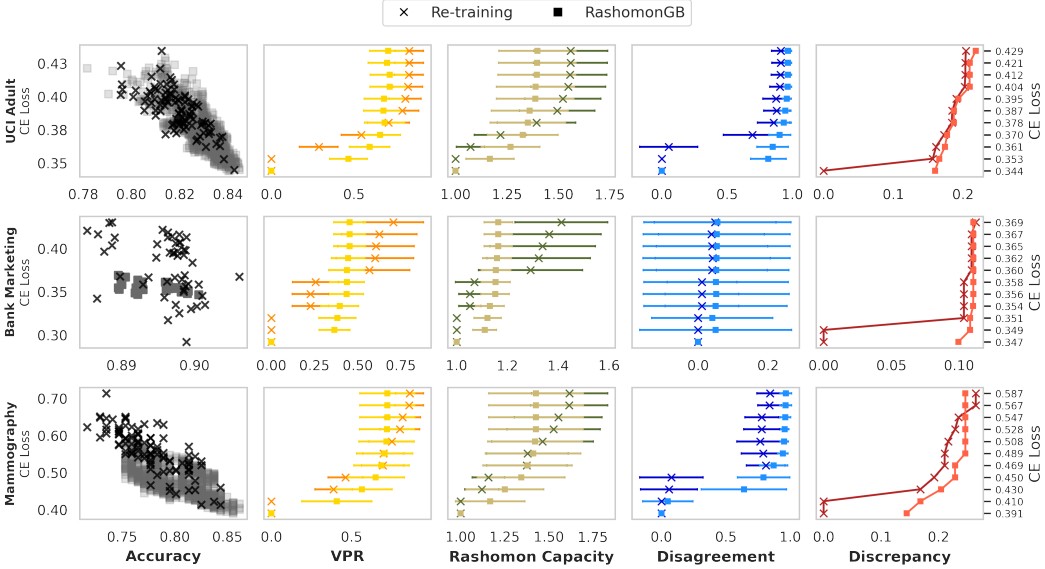

Figure E.8: `Re-training` vs. `RashomonGB` in exploring the Rashomon set for predictive multiplicity metrics estimation. In the leftmost column, each marker represents a model. The rightmost 4 figures in a row share the same y-axis for the loss difference (values shown at the right), i.e., $L_{P_S}(h^*) + \epsilon$ in Eq. (1). Higher predictive multiplicity values mean a better estimate.

### E.3 Computational time comparison

We compare the training time, and the time to obtained one model for `re-training` and `RashomonGB` with $T = 10$, $m = 10$, $\alpha = 0.8$ and depth-2 decision tree as weak learners. We repeated the experiments 3 times with different random seeds and report the time (in seconds) in Table E.4. Note that `re-training` and `RashomonGB` share the same training time, and the values we show for the columns `re-training` and `RashomonGB` are the inference time. It is clear that the time cost to obtain a model from `RashomonGB` is consistently and significantly smaller than that from `re-training`. All the runtimes we reported here are computed on the same machine in an Amazon EC2 g4dn.8xlarge instance.

Table E.4: Comparisons of training time, and the time to obtained a model for `re-training` and `RashomonGB`. All values are in seconds, and are repeated with 3 experiments with different seeds.

| Datasets | Training | `retraining` per model | `RashomonGB` per model |
|---|---|---|---|
| UCI Adult | $2.6667 \pm 0.4714$ | $0.0010 \pm 0.0000$ | $0.0000 \pm 0.0000$ |
| ACS Income | $54.6667 \pm 0.4714$ | $0.4000 \pm 0.0000$ | $0.0273 \pm 0.0000$ |
| Bank Marketing | $2.6667 \pm 0.4714$ | $0.0333 \pm 0.0471$ | $0.0013 \pm 0.0005$ |
| Mammography | $0.0000 \pm 0.0000$ | $0.0010 \pm 0.0000$ | $0.0000 \pm 0.0000$ |
| Contraception | $0.0000 \pm 0.0000$ | $0.0667 \pm 0.0471$ | $0.0003 \pm 0.0005$ |
| Credit Card | $0.6667 \pm 0.4714$ | $0.0007 \pm 0.0005$ | $0.0000 \pm 0.0000$ |
| COMPAS | $0.3333 \pm 0.4714$ | $0.0667 \pm 0.0471$ | $0.0003 \pm 0.0005$ |

### E.4 Supporting experiments for Section 3.3

We validate our discussion in Section 3.3 that more boosting iterations could lead to a larger Rashomon set. We perform `RashomonGB` on the UCI Adult dataset with $m = 100$, $\alpha = 0.8$, and decision trees as weak learners with different numbers of boosting iterations $T = [1, \cdots, 10]$ in Figure E.9. On the left side, we show the CE Loss vs. accuracy of the models in each iteration. It is clear that the CE losses decreases and the accuracy increases with more boost iterations. On the right side, we show the percentage of models in the Rashomon set in each boosting iteration under different loss constraints. For example, when the loss constraint is around $0.40$, more than $90\%$ of the models in iteration $T = 10$ are in the Rashomon set but only $50\%$ of the models in iteration $T = 4$.

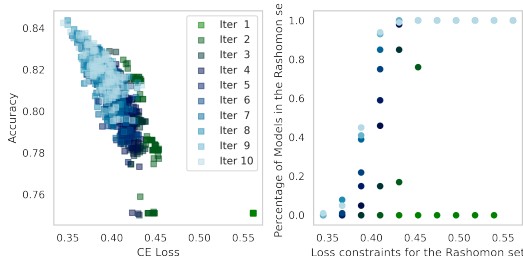

Figure E.9: Fraction of models in the Rashomon set in different number of boosting iterations $T$ and CE loss constraints.

## E.5 Ablation study on different types of weak learners

We compare `re-training` and `RashomonGB` on the UCI Adult dataset with $T = 10$, $m = 10$, $\alpha = 0.8$, and different weak learners (depth-2 decision trees, feedforward neural networks (one-hidden layer with 10 neurons, RelU activations, Adam optimizer, trained with 10 epochs with a constant learning rate 0.001), and linear regression) in Figure E.10. `RashomonGB` outperforms `re-training` on decision tree and neural network regressors as these two types of weak learners are more complicated. Linear regression is too simple and therefore `RashomonGB` is not able to find diverse models in the Rashomon set. Neural network weak learners achieves the highest accuracy (1% higher) than linear regression and decision trees.

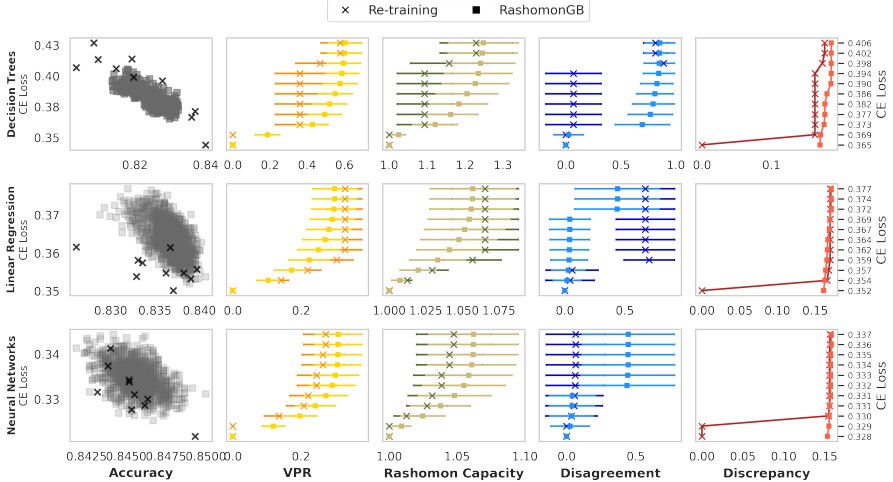

Figure E.10: `Re-training` vs. `RashomonGB` in exploring the Rashomon set for predictive multiplicity metrics estimation with different weak learners.

## E.6 Ablation study on different depths of decision tree regressors

We compare `re-training` and `RashomonGB` on the UCI Adult dataset with $T = 10$, $m = 10$, $\alpha = 0.8$, and decision trees of different depths as weak learners in Figure E.11. As the tree depth increases, i.e., weak learners are more complicated, and both `re-training` and `RashomonGB` achieve higher accuracy and lower CE losses. For the same loss constraints, e.g., CE loss $\approx 0.370$, a more complicated weak learner will lead to more predictive multiplicity compared to simpler weak learners, since the hypothesis space is larger, leading to a larger Rashomon set.

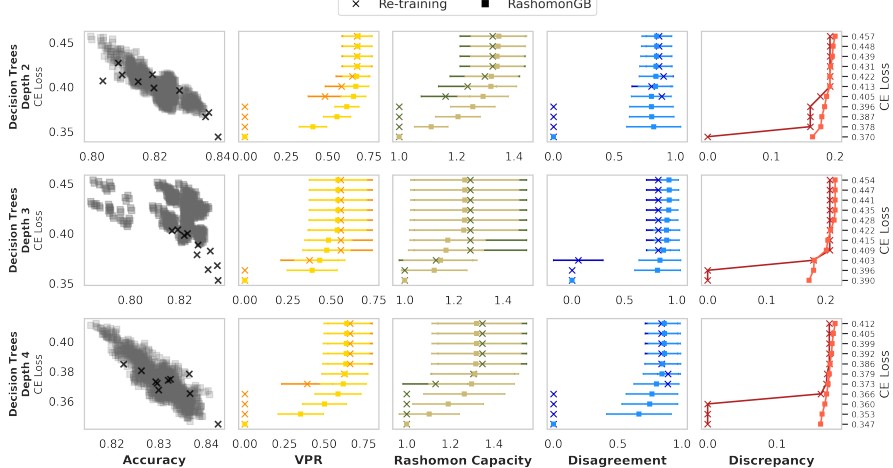

Figure E.11: `Re-training` vs. `RashomonGB` in exploring the Rashomon set for predictive multiplicity metrics estimation with decision tree regressors of different depths as weak learners.

### E.7  Ablation study on number of boosting iterations

We compare `re-training` and `RashomonGB` on the UCI Adult dataset with $m = 10$, $\alpha = 0.8$, and decision trees as weak learners with different numbers of boosting iterations $T = [2, 5, 10]$ in Figure E.12. For $T = 5$, `RashomonGB` finds models with higher losses and does not perform as good as `re-training`. For $T = 2$ and $T = 10$, `RashomonGB` outperforms `re-training` as usual. Note that for the same size of Rashomon set with CE losses $\approx 0.370$, more boosting iterations ($T = 10$ vs. $T = 2$) could lead to a larger effect of predictive multiplicity, as discussed in Section 3.3.

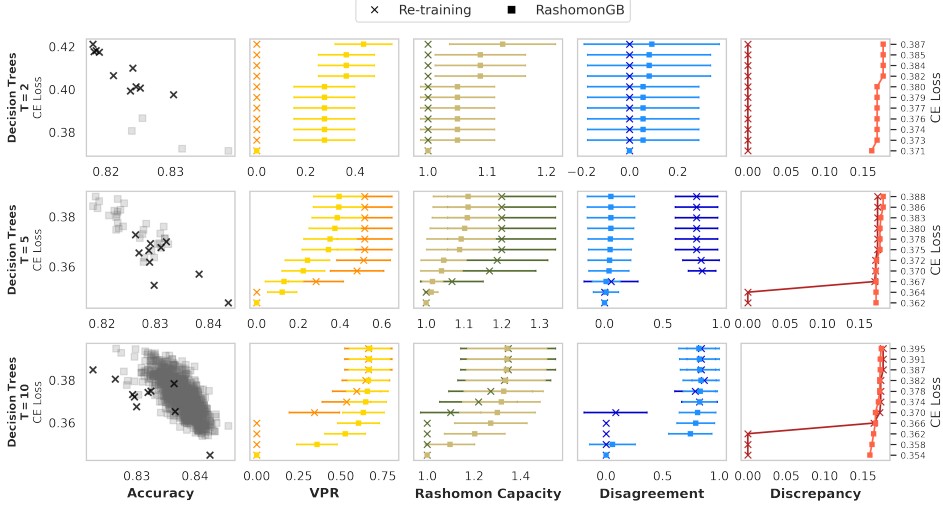

Figure E.12: `Re-training` vs. `RashomonGB` in exploring the Rashomon set for predictive multiplicity metrics estimation with different boosting iterations $T$.

### E.8  Ablation study on the number of model in each iteration

We compare `re-training` and `RashomonGB` on the UCI Adult dataset with $T = 10$, $\alpha = 0.8$, decision trees as weak learners, and different number of model in each iteration $m = [2, 5, 10]$ in Figure E.13. When $m = 2$, `re-training` only obtains 2 models whereas `RashomonGB` is still able to find more than 1000 models in the Rashomon set. Therefore, when $m$ is smaller, i.e., less budget on computational resources, `RashomonGB` consistently performs better than `re-training`.

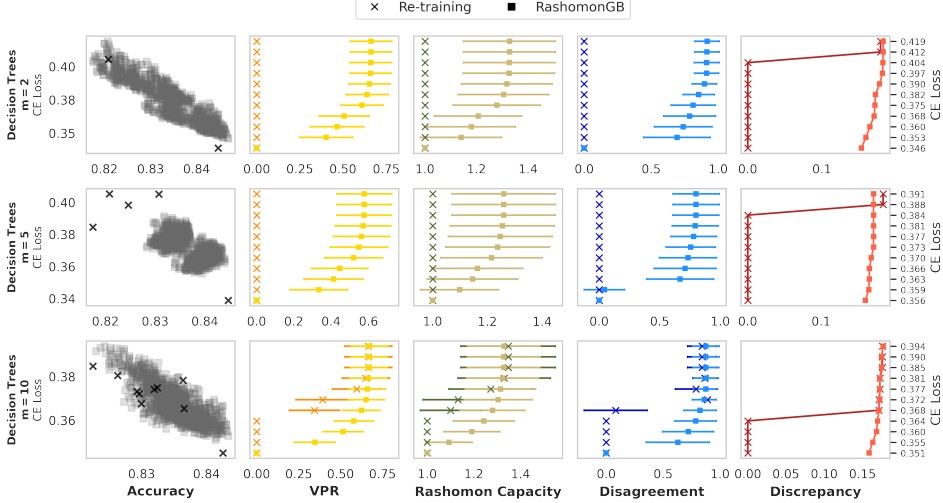

Figure E.13: `Re-training` vs. `RashomonGB` in exploring the Rashomon set for predictive multiplicity metrics estimation with different number of models $m$ in each iteration.

## E.9 Comparison with predictive uncertainty estimation in gradient boosting

Prediction uncertainty [35, 2] indeed differs fundamentally from predictive multiplicity. Prediction uncertainty, derived from a Bayesian perspective, seeks to reconstruct the distribution $p(Y|x)$ for a given sample $x$ and assess metrics such as variance or negative log-likelihood of $Y$, typically involving only one model without specific loss constraints. Conversely, predictive multiplicity involves evaluating multiple models within the Rashomon set that exhibit similar loss, thereby reflecting a variety of potential outcomes for the same inputs. To elucidate these distinctions, we have compared our re-training strategy, RashomonGB, with the prediction uncertainty methods—NGBoost [27], PGBM [70], and IBUG [13]—using the UCI Contraception dataset. For a rigorous comparison, in Figure E.14, we applied these prediction uncertainty methods to estimate $p(Y|x)$ (parameterized as Gaussian), sampled 1024 values of $y$, and computed the corresponding Rashomon set and predictive multiplicity metrics. The results demonstrate that RashomonGB encompasses the widest range of models, thereby providing consistently higher and more robust estimates of predictive multiplicity metrics. This comparison highlights the unique capabilities of RashomonGB in capturing a broader spectrum of potential model behaviors within the dataset.

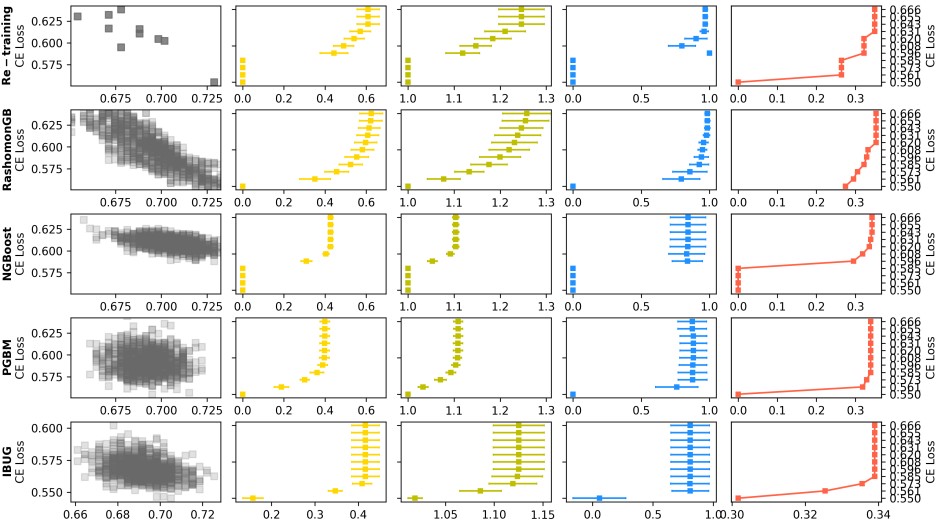

Figure E.14: `Re-training` (1st row) vs. `RashomonGB` (2nd row) vs. prediction uncertainty methods (`NGBoost` (3rd row), `PGBM` (4th row), and `IBUG` (5th row)) in exploring the Rashomon set for predictive multiplicity metrics estimation on the UCI Contraception dataset. For the prediction uncertainty methods, we following the assumption of Gaussianity of $P(Y|x)$ and estimate the mean and covariance from `NGBoost`, `PGBM`, and `IBUG`. Using the estimated mean $\theta$ and covariance $\Sigma$, we sample 1024 $y$ from the distribution $P(Y|x) \approx \mathcal{N}(\theta, \Sigma)$. In the leftmost column, each marker represents a model for `Re-training` and `RashomonGB`, or a realization of $Y$ sampled from prediction uncertainty methods . The rightmost 4 figures in a row share the same y-axis for the loss difference (values shown at the right), i.e., $L_{P_S}(f_T^*) + T\epsilon$. Higher predictive multiplicity values mean a better estimate. For the leftmost column, it is clear that `RashomonGB` covers a widest range of models, leading to consistently higher estimates of VPR and the Rashomon Capacity. For decision-based metrics such as disagreement, `RashomonGB` has the better estimate when CE Loss is higher; however, for discrepancy `IBUG` has the better estimate.

## E.10 Hyperparameter sensitivity results related to mitigation methods in Section 4.3

Figure E.15 illustrates the effect of model hyperparameters on our two measures of interest: 0-disagreement and accuracy. We report mean of each measure over 20 random train/test splits. We evaluate several datasets with larger $\Delta$0-disagreement in Figure 5. In (a), varying the ensemble size $E$ tends to reduce disagreement on the 5 datasets evaluated. From this result, we use $E = 20$ when fixing $E$ in our experiments. In (b), 0-disagreement increases over increasing $k$. We select the $k$ weak learners from the candidate set in ascending order of loss. Therefore, increasing $k$ tends to add models of decreasing quality. Furthermore, the 0-disagreement measure is strict; it requiring *every* additional model to have the same prediction on the sample. In (c), 0-disagreement tends to decrease to around $\lambda = 5$. From this result, we tune on $\lambda \in [0, ..., 5]$ in our experiments. Finally, the bottom row demonstrates that accuracy is *not* sensitive to changes in any of these hyperparameters.

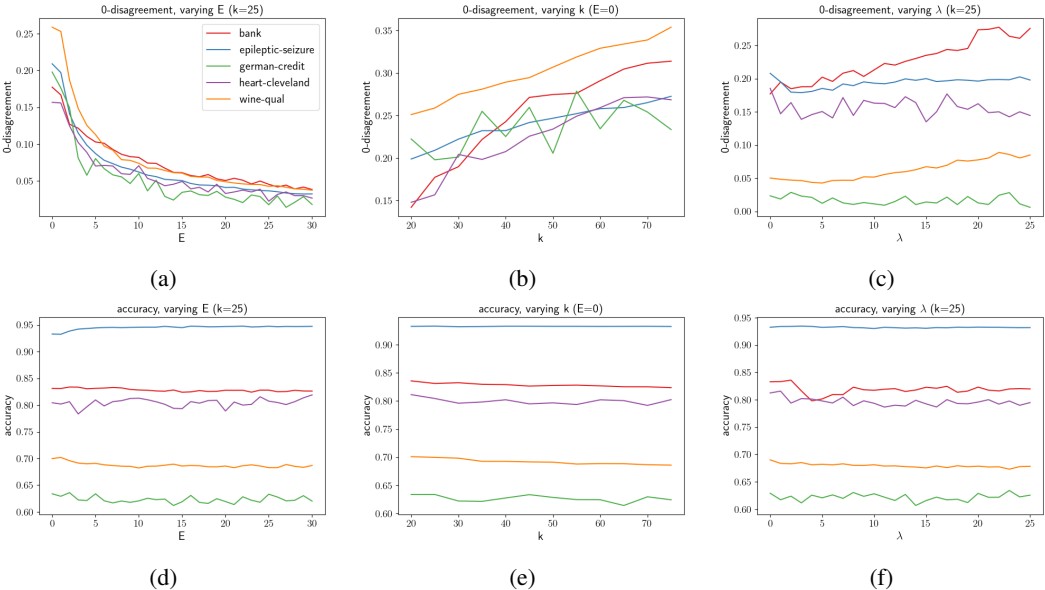

Figure E.15: Comparison of varying hyperparameters of the ensemble size ($E$) the loss reweighting penalty ($\lambda$), and the total number of trained models $k$. The first row reports the 0-disagreement (*Lower is better*), the second reports the accuracy. We report average statistics over 20 random train-test splits.

