# OpenReview forum: "RashomonGB: Analyzing the Rashomon Effect and Mitigating Predictive Multiplicity in Gradient Boosting"
_NeurIPS.cc/2024/Conference — NeurIPS 2024 poster_

### Official Review · Reviewer_E8Ec · 2024-07-08

**Soundness:** 2
**Presentation:** 3
**Contribution:** 3
**Rating:** 6
**Confidence:** 4

**Summary:**

This paper proposes a method (RashomonGB ) to estimate the Rashomon sets/predictive multiplicity of gradient boosting models. It estimates multiple ($m$) models at each stage (effectively performing a local exploration) and then combine all such models in the end to construct $m^T$ models for Rashomon set computation, where $T$ is the number of iterations of the boosting. On several datasets the paper shows that RashomonGB performs better than re-training with $m$ seeds, in that at the fix $\epsilon$ (loss difference) level, RashomonGB tends to show more predictive multiplicity.

**Strengths:**

Predictive multiplicity is an important topic. The paper is generally clear and well-written. The proposed method is a sensible first method for boosting algorithms, which was previously underexplored. I think the proposed method is likely adopted by people who care about this problem as it's intuitive and easy to implement.

**Weaknesses:**

1. The current exploration strategy is fast to compute, but I'm not sure if this follows the motivation of Rashomon set very well. While the authors mention one example on the Contraception dataset where re-training underestimates the predictive multiplicity, in general RashomonGB might create models that are more correlated than normal (because the "backbone" is the same GB model), thus underestimating the predictive multiplicity. Right now, the conclusion shows otherwise probably because the number of re-training is too small.

2. Regarding the experiment, if I read this correctly, currently we use more compute for RashomonGB as well (by combining different weak models), so it is also not quite a fair comparison in my opinion. I would be very interested to see some estimate of how much compute RashomonGB saves against re-training, by running more re-training and see when are the metrics in Fig3 in the two methods become comparable.



minor: one "RashomonGB" in L290 should be "re-training".

**Questions:**

1. What's $\epsilon_{t_1}$ (and $\epsilon_{t_2}$) in L243-L244? Isn't epsilon a quantity set by the user?


2. In L282-283, do we construct 10 final models and 1024 for re-training and RashomonGB, respectively? If only 2 out of $m$ models are used why train $m$ of them (L282-283) for RashomonGB?

3. Related to the above, I originally thought there is a model "filtering" step in each iteration $t$, and wonder how $\epsilon_t$ is set for each iteration. However, from L282-283 it seems like we just randomly pick a few models and brute-force combine all weak models for the final Rashomon set exploration. Could the authors clarify?

4. Are Fig 4 measured on the test set? If so, then it's not clear how useful this is as we cannot choose models basing on test performance - did the authors try picking models on the frontier basing on the validation set and then plot this on the test set? Right now, due to the sheer number of final models generated by RashomonGB, it's unclear if the models with better trade-off are just lucky.

---

> ### Author Rebuttal · Authors · 2024-08-06
>
> We appreciate the reviewer's constructive feedback and encouragement. Below, we systematically address each weakness and question raised in the review.
>
> For Weakness 1, the estimates in Figure 3, derived from both re-training and RashomonGB, utilized the same training cost. Specifically, each method required training T\*m = 10\*10 = 100 weak learners (except CIFAR-10; cf. Line 282). The re-training approach, using different random seeds, explores the Rashomon set in a "global" fashion, potentially yielding a more diverse set of models within this space. Conversely, RashomonGB explores the set in a more "local" manner, focusing on models that incorporate the same weak learners. Despite this, under identical training costs, RashomonGB can explore exponentially more models than the re-training strategy. It's important to note that for complex hypothesis spaces, like Gradient Boosting (GB) used here, any method, including re-training, will likely underestimate predictive multiplicity due to the sheer computational impracticality of fully exploring the Rashomon set. This introduces an inevitable trade-off between the efficiency (training cost) and the effectiveness (degree of underestimation) in estimating predictive multiplicity. In Figure 3, our goal was to highlight the efficiency and effectiveness of RashomonGB under equal training costs. However, as the number of re-training increases—thereby increasing the training cost—the re-training method may surpass RashomonGB in effectively estimating the diversity within the Rashomon set.
>
> Regarding Weakness 2, although the computational costs for re-training and RashomonGB are equivalent, RashomonGB significantly reduces the time required to obtain a model. This is because RashomonGB explores an exponential search space, making it much more efficient. This efficiency is compellingly demonstrated in the comparative analysis of computational times shown in Table E.4 of Appendix E.2, where RashomonGB's model generation speed significantly outpaces that of re-training. For instance, for the ACSIncome dataset, both methods recorded a training time of 54.67 seconds. However, while the inference time per model for re-training is 0.4 seconds, it is only 0.02 seconds for RashomonGB. This means that, under the same training cost, RashomonGB is 20 times more efficient in terms of generating models from the Rashomon set compared to the re-training strategy. We will address this comment by clearly directing readers in the revised main text to the additional experiments detailed in the Appendix. This will help ensure that the relevant information is easily accessible and comprehensible.
>
> For Question 1, indeed, $\epsilon$ is a parameter configured by the user. As stated in Lines 243-244, maintaining the same $\rho$—defined as the probability in Proposition 1—results in more iterations increasing the conditional mutual information, which in turn leads to a larger $\epsilon$. This insight serves as a practical guideline advising users against choosing smaller values of $\epsilon$ in subsequent boosting iterations.
> This effect is further illustrated in the Ablation study detailed in Appendix E.3. Figure E.8 demonstrates this by fixing $\epsilon$ for each iteration, re-training with different random seeds, and computing the percentage of models (i.e., $\rho$ in Proposition 1) in the Rashomon set for each iteration. It is observable that, with the same percentage (i.e., the same $1-\rho$), $\epsilon$ increases. This observation reinforces the results suggested by Proposition 3, validating the relationship between $\epsilon$ and $\rho$ under consistent conditions.
>
> For Question 2, indeed, we implemented the re-training strategy using 10 different random seeds for the Gradient Boosting with 10 iterations. This strategy involved training 100 weak learners (10 per iteration). Ideally, RashomonGB can generate up to $10^{10}$ final models. However, as noted in footnote 5, even selecting 3 models per iteration for RashomonGB would yield over 59,000 final models, which exceeds our storage capabilities. Consequently, we chose to select 2 models per iteration for RashomonGB. Comparatively, if we train only 2 models in each iteration, the re-training strategy results in just 2 final models, whereas RashomonGB still generates 1024 models. This underscores the superior efficiency of RashomonGB in generating a higher number of models under the same training cost, thereby providing a broader exploration of the model space.
>
> For Question 3, indeed, there is a filtering process in place that screens out models with an MSE loss greater than 0.1 (i.e., $\epsilon_t = 0.1$) and retains models with an MSE loss smaller than 0.01 until $m=10$ models are collected at each iteration.
>
> For Question 4, the models situated at the accuracy-fairness trade-off frontier were selected using a hold-out validation set post-training, and the results depicted in Figure 4 were evaluated using the test set. Compared to the re-training strategy, the lower cost of obtaining models through RashomonGB enhances the likelihood of identifying a model with a more optimal operation point. The advantage of RashomonGB becomes even more pronounced when dealing with larger datasets. In such scenarios, re-training and re-training-based fairness intervention algorithms, such as Reduction (and FaiRS) and Rejection, may incur significantly higher training costs.
>
> We will add the additional explanation and results, and fix the typo (e.g., "RashomonGB" in L290) in the revision. Thanks again and we would be happy to provide more clarifications and answer any follow-up questions.

---

### Official Review · Reviewer_XEPy · 2024-07-13

**Soundness:** 3
**Presentation:** 2
**Contribution:** 3
**Rating:** 5
**Confidence:** 2

**Summary:**

This paper presents an approach that compute Rashomon set for gradient boosting algorithm where the set can be obtained through products over weak learners at each step rather than sampling them through retraining. The authors further proposed a dataset related Rashomon bound through sub-Gaussian assumption, where mutual information between hypothesis space and dataset shows the predictive multiplicity, which can further decomposed into model uncertainty and quality of data. Experiments show the proposed solution offers more models in Rashomon set than retraining given the same computation budget.

**Strengths:**

The rough idea of the proposed approach is straightforward since decomposing Rashomon set search on boosting algorithm can be a "standard" operation given the unique residual learning property of boosting algorithms. The novelty of the proposed approach is probably more from "our work is the first to explore the Rashomon effect for gradient boosting".

The dataset related Rashomon set bound seems an interesting point. But it needs some justification for the key assumption of it (sub-Gaussian). Proposition 2 seems make sense given the positive relation between number of boosting iterations and Rashomon set (also for dataset size).

Experiments in 4.2 seem interesting. I would love to see more experiments like it.

**Weaknesses:**

I got some difficult time to understand the introduction and abstract of this paper even I have read some literatures about Rashomon effect and predictive multiplicity. It is simply hard to read given the narrative there. Especially the second paragraph of introduction; it gets me confused and self-questioning my understanding of Rashomon effect from other works.

**Questions:**

Why boosting algorithms?
Further justification about the dataset related Rashomon set bound?

**Limitations:**

No hard limitation I can see.

---

> ### Author Rebuttal · Authors · 2024-08-06
>
> We thank the reviewer for the feedback! We clarify the weakness and answer the reviewer's question below.
>
> To address the weakness pointed out, it would be helpful if the reviewer could specify which parts of the second paragraph in the Introduction are unclear or difficult to understand during the author-reviewer discussion phase. To enhance clarity, we will expand on the concepts introduced in both the Abstract and the Introduction.
> In this paper, our objective is to explore both the positive (e.g., improved model selection as discussed in Section 4.2) and negative (e.g., predictive multiplicity as detailed in Section 4.1) impacts of the Rashomon effect using gradient boosting algorithms. These algorithms uniquely employ a sequential training procedure that focuses on learning the residuals of the data.
> The Rashomon effect articulates that within a given hypothesis space, numerous models can achieve similar performance levels (such as 99\% accuracy). These similarly performing models can be grouped into what is known as the Rashomon set. The positive aspect of the Rashomon effect is particularly significant in tasks associated with responsible machine learning, which often requires models to possess additional properties (such as group fairness) without significantly sacrificing performance. In essence, the pursuit of responsible machine learning is about finding models within the Rashomon set that meet these extra constraints (as exemplified by the fairness considerations in Section 4.2). The Rashomon effect thus provides assurance of finding viable solutions when optimizing for responsible machine learning goals.
> Conversely, the negative aspect, termed predictive multiplicity, occurs when a model selected at random from the Rashomon set leads to inconsistent decisions for some individuals (e.g., affecting 5\% of the samples as illustrated in Figure 3 of Section 4.1). This unpredictability can undermine the reliability of the machine learning process.
> By elaborating on these concepts, we aim to resolve any confusion and reinforce the significance of our investigation into the dual implications of the Rashomon effect within gradient boosting frameworks.
>
> In response to the Question, boosting algorithms are prevalently utilized for tabular datasets, particularly in the realm of trustworthy machine learning, such as fairness interventions detailed in Section 4.2. Notably, boosting algorithms have been shown to surpass deep learning on tabular datasets, as referenced in [35]. Despite this, existing theoretical frameworks that explore the Rashomon effect and predictive multiplicity have primarily focused on linear classifiers [55, 72], generalized additive models [15], sparse decision trees [74], and neural networks [42]. The unique sequential training procedure of boosting and its influence on the characterization of the Rashomon set remain poorly understood. This paper aims to bridge this gap, providing mathematical tools that could also extend to other sequential residual training schemes. We initially discuss our motivation for focusing on boosting algorithms in Lines 75-90 and 117-127. In the revision, we will reorganize this content to better highlight our motivation, as suggested by the reviewer.
>
> Regarding the assumption of sub-Gaussianity of the loss function, this represents a generalization beyond mere boundedness. While it is feasible to assume boundedness of the loss function—a common and practical approach readily achieved by clipping the loss, as mentioned in Lines 185-187—we opt for the sub-Gaussian assumption in Proposition 1 to allow for a broader analysis. This paper emphasizes the novelty of dataset-related bounds on the Rashomon set, where previous studies have largely concentrated on the hypothesis space. This perspective underscores our novel contribution to the understanding of the Rashomon effect and predictive multiplicity in machine learning.
>
> We will include additional explanations regarding the sub-Gaussian assumption in the revised Section 3. We welcome any follow-up questions and are happy to provide further clarification. Thank you!

---

### Official Review · Reviewer_JGsC · 2024-07-13

**Soundness:** 3
**Presentation:** 3
**Contribution:** 3
**Rating:** 7
**Confidence:** 4

**Summary:**

The paper studies the Rashomon effect in gradient boosting, a commonly used algorithm for tabular datasets, but something that has not received enough attention in multiplicity literature. The paper provides several theoretical discussions on the size of the Rashomon set and the impact of the number of iterations on multiplicity in GBRTs. Furthermore, the paper proposes RashomonGB, a method to create an exponential number of ‘near-optimal models’ by training only a polynomial number of models. With more models in the Rashomon set, the use of RashomonGB can create several downstream benefits without any extra cost of training, shown empirically by the authors.

**Strengths:**

- Multiplicity in GBRTs, or generally any gradient-boosting algorithm, has not been studied in the literature, and so the authors provided a novel discussion, especially given the importance of these algorithms in tabular settings.
- The paper provides several theoretical discussions backed by empirical support. The insights on the growing Rashomon set with iterations were quite interesting, although I have concerns about the validity of these insights (see Weaknesses).
- Multiplicity quantification can be quite costly, and various methods in pursuit of reducing this cost can significantly benefit further auditing. The use of RashomonGB, as proposed by the authors, can be an important step in that direction for gradient-boosted algorithms.

**Weaknesses:**

- While the presentation of the rest of the concepts and the theoretical discussion were easy to follow, important details about the RashomonGB method and the details of the empirical setup were either missing (even from the Appendix) or imprecise. For instance, the Rashomon set of the gradient boosting algorithm isn’t going to simply be the iterative extension of Rashomon sets at every residual level, i.e., equation 4 is imprecise. Similarly, it seems that the epsilon value of the Rashomon set increases with more iterations, and thus it is confusing to me whether the insight that more iterations create bigger Rashomon sets is a result of multiple iterations or simply a result of bigger epsilon. See the section ‘Questions’ for more detailed comments and some follow-up questions. Edit after rebuttal: Acknowledged, correct and clarified.
- There are other methods to measure predictive uncertainty in gradient-boosted algorithms. Some examples based on a cursory search (there might be more, as I’m not too familiar with GBRTs) - https://arxiv.org/abs/2205.11412 https://arxiv.org/pdf/1910.03225 https://arxiv.org/abs/2106.01682 - While I understand that prediction uncertainty is not the same as predictive multiplicity, the two are closely related, and when proposing a better method to measure multiplicity, the paper should compare itself with other stronger baselines than just retraining. Just as previous works have proposed using Monte Carlo Dropout (which was initially created as a method to measure uncertainty) as a measure of multiplicity, uncertainty measurement baselines for GBRTs could have been adopted to create reasonable baselines, and would have made the results a lot stronger. Edit after rebuttal: Acknowledged and added.

**Questions:**

My questions and comments mostly revolve around the RashomonGB formulation.
- I don’t believe equation 4 is correct. A model formed from residual models that are present in their Rashomon sets at every step does not necessarily make a model that will be present in the Rashomon set overall. That’s because the composition of GBRTs occurs at the prediction level, while Rashomon sets are defined by the authors at the loss level. Equation 4 probably would have been true if the loss function had a linear relationship with the model predictions, which is not an assumption I see being made anywhere in the paper. This also makes me question the empirical results, because if the RashomonGB formulation isn’t precise, do the models across which the authors calculate multiplicity even belong to the same Rashomon set? Edit after rebuttal: Acknowledged and corrected.
- Can the authors comment on why they compare two situations with different Rashomon parameters and make claims on their multiplicity? For example, Proposition 3 and the following paragraph. A Rashomon set would of course be bigger with a larger value of epsilon, and having that variability when talking about other trends doesn’t seem convincing to me. Edit after rebuttal: Confusion clarified.
- What was the exact epsilon value used for the experiment? I couldn’t find it anywhere in the paper. Moreover, I hope that given the Rashomon sets for the RashomonGB setup were defined with T*epsilon as the new epsilon value, the same freedom was also given to retraining. Again, if the comparison was done across methods with different epsilon values (which might not be the case, but I don’t know the details), that does not make sense to me. Edit after rebuttal: Appropriate information added.

**Limitations:**

- A central piece of the paper is their method RashomonGB. While the authors do try to emphasize the importance of this method by highlighting the number of models that can be created using their method, just the number alone is not enough to imply a better method for measuring multiplicity. Even assuming that the comparisons are indeed fair (see Questions), the differences in multiplicity are not very severe, and that makes me wonder if combining pieces of various residual models actually gives us new interesting models or do we just end up with similar models as already seen during retraining. The authors acknowledge this briefly in their limitations paragraph. Edit after rebuttal: Appropriate details added and clarified.

---

> ### Author Rebuttal · Authors · 2024-08-06
>
> We appreciate the reviewer’s constructive feedback. We address the weaknesses, questions, and limitations point-by-point below.
>
> For Weakness 1, please refer to our responses to Question 1 and Question 2.
>
> For Weakness 2: as indicated, prediction uncertainty indeed differs fundamentally from predictive multiplicity. Prediction uncertainty, derived from a Bayesian perspective, seeks to reconstruct the distribution $p(Y|x)$ for a given sample $x$ and assess metrics such as variance or negative log-likelihood of $Y$, typically involving only one model without specific loss constraints. Conversely, predictive multiplicity involves evaluating multiple models within the Rashomon set that exhibit similar loss, thereby reflecting a variety of potential outcomes for the same inputs. To elucidate these distinctions, we have compared our re-training strategy, RashomonGB, with the prediction uncertainty methods cited by the reviewer—NGBoost [R1], PGBM [R2], and IBUG [R3]—using the UCI Contraception dataset. This comparison is detailed in Figure R.2 of the attached one-page PDF. For a rigorous comparison, we applied these prediction uncertainty methods to estimate $p(Y|x)$ (parameterized as Gaussian), sampled 1024 values of $y$, and computed the corresponding Rashomon set and predictive multiplicity metrics. The results demonstrate that RashomonGB encompasses the widest range of models, thereby providing consistently higher and more robust estimates of predictive multiplicity metrics. This comparison highlights the unique capabilities of RashomonGB in capturing a broader spectrum of potential model behaviors within the dataset.
>
> For Question 1: when utilizing GBRT for classification tasks, the method actually performs a regression on the log-likelihood using the MSE loss (Lines 140-143). The MSE loss qualifies as sub-Gaussian, aligning with the assumption set up in Proposition 1. Additionally, the pseudo-residual of the MSE loss exhibits a linear relationship between the prediction and the output of each iteration (Line 597). This allows us to apply Proposition 1 to aggregate the losses across iterations, leading to the formulation of Proposition 2, which is a detailed extension of Equation 4.
> Equation 4 delivers the concept of constructing the overall Rashomon set by the Rashomon sets in each iteration. We thank the reviewer for noting the error and will change the equality to $\supseteq$.
> We will ensure that these assumptions and their implications are clearly articulated in the revised manuscript to eliminate any ambiguity.
>
> For Question 2, we thank the reviewer for highlighting the potential confusion regarding the direction of reasoning related to $\epsilon$. To clarify, we do not start with the assumption that $\epsilon$ increases with each iteration; rather, this conclusion emerges from Proposition 3. Proposition 3 and the discussions in Section 3.3 indicate that with a constant $\rho$—as defined in Proposition 1—additional iterations result in increased conditional mutual information, which in turn necessitates a larger $\epsilon$, as detailed in Line 243. This dynamic is visually supported by Figure 2, where the conditional entropy—and consequently the mutual information—escalates as the boosting process progresses. This is because the Rashomon effect accumulates over the sequential learning problems tackled in each iteration, emphasizing the cumulative impact on the diversity within the model space. The Ablation study in Appendix E.3 further clarifies the selection of $\epsilon$ through its iterations. Figure E.8 demonstrates that fixing $\epsilon$ while re-training with different random seeds results in a decreasing percentage of models ($\rho$ from Proposition 1) in the Rashomon set. This implies that to maintain a consistent $\rho$, the chosen $\epsilon$ must increase. This observation corroborates Proposition 3's findings on the relationship between $\epsilon$ and $\rho$.
>
> For Question 3, for the experiments of reporting predictive multiplicity in Section 4.1, we report the Rashomon parameter $\epsilon$ in the vertical axis (leftmost column) in Figure 3. For the experiments of fair model selection, we report the $\epsilon$ in the caption of Figure 4. For the experiments of mitigating predictive multiplicity by model averaging, we report the $\epsilon$ (in terms of the improvement of accuracy) in the vertical axis in Figure 5. Note that the $\epsilon$ we report here is the overall Rashomon parameter after $T = 10$ iterations, i.e., T\*$\epsilon$. Moreover, we do not compare models with different $\epsilon$ as it is clearly unfair. We provide an explanation on how to interpret the results of Figure 3 in Figure R.1 in the one-page PDF, please check!
>
> Finally for the limitation, indeed, as discussed in Section 5, while re-training with different random seeds offers a "global" exploration of models within the Rashomon set, RashomonGB conducts a "local" exploration. However, RashomonGB demonstrates greater efficiency than the re-training strategy, highlighting a trade-off between the efficiency of exploring the Rashomon set and the effectiveness of capturing model diversity. For instance, in Figure 3, numerous instances across different datasets show that the re-training strategy significantly underestimates predictive multiplicity metrics (e.g., VPR and Rashomon Capacity are reported as zero), particularly when $\epsilon$ is small. This underestimation often occurs because re-training fails to gather a sufficient number of models. To the best of the authors' knowledge, finding the (sub-)optimal strategy for exploring the Rashomon set remains an active area of research. We will incorporate this additional discussion into the revised Section 5 to provide a comprehensive understanding of the trade-offs involved and the current state of research in this field.
>
> Thanks again and we would be happy to provide more clarifications and answer any follow-up questions.

---

> > ### Comment · Reviewer_JGsC · 2024-08-09
> >
> > Thank you for your response. Some of my concerns have been answered and I will raise my scores. However, I still have some followup questions for other concerns.
> >
> > > Question 2
> >
> > Consider I'm a user who wants to deploy these GBRTs. I would have some $\epsilon$ value in mind before I begin, eg, I can allow a 0.1 difference in loss if it allows me benefits elsewhere, say choosing fairer models. I train multiple GBRTs for 50 iterations each and get a set of models. I then filter them to find 'good models', i.e., those in my Rashomon set and would use them to calculate multiplicity, choose the model with the best fairness scores, etc.
> >
> > Now instead of training for 50 iterations, what if I had trained for 100 iterations? What I don't understand is why would I change my threshold ($\epsilon$) to 0.2. Wouldn't I still only want to find benefits while making sure I'm just a 0.1 loss difference away from the best model? Maybe a different way to interpret this could be that under the fixed $\epsilon$, I'm less likely to get models (as noted by the authors that the same $\epsilon$ means smaller $\rho$). In other words, I'd have less number of models to choose from if I forced the same $\epsilon$ threshold. But I feel this is a more realistic setting, and in this case, my Rashomon set has shrunk, not grown!
> >
> > One thing to note, the size of the Rashomon set is NOT a proxy for multiplicity. Thus, despite being a smaller set, this set of models can still have higher multiplicity, a bigger range of coverage in terms of fairness, and so on. So all the following results can, in principle, still exist. And of course, they do exist. But the narrative of increasing Rashomon set size is bothering me.
> >
> > I may be missing or misinterpreting something. Happy to hear more clarification from the authors.
> >
> > > Question 3
> >
> > Ahh, I see, now the results make more sense. This was confusing at first. Please make sure to add the clarification and proper explanation on how to interpret the figure in the final version.

---

> ### Author Response · Authors · 2024-08-09
> **Further response to Reviewer JGsC's comments**
>
> We appreciate the additional feedback from the reviewer!
> Consider a scenario where we train $m$ models per iteration for $T\_1$ iterations, and subsequently extend the training up to $T\_2$ iterations, where $T\_1 < T\_2$.
> We can construct the overall Rashomon sets using models obtained from both the $T\_1$-th and $T\_2$-th iterations.
> With the same threshold $\epsilon$ for the Rashomon set, we can infer from Propositions 2 and 3 that the probability $1-\rho$ of a model belonging to the Rashomon set will decrease over the iteration.
> Let $\rho\_1$ and $\rho\_2$ represent the probabilities for iterations $T\_1$ and $T\_2$, respectively, then $1-\rho\_1 \geq 1-\rho\_2$.
> Consequently, the number of models from the $T\_1$-th iteration that are included in the Rashomon set with threshold $\epsilon$ will be $m^{T\_1} \times (1-\rho\_1)$. Similarly, for the $T\_2$-th iteration, the count will be $m^{T\_2} \times (1-\rho\_2)$.
> It is important to note that although $1-\rho\_1 \geq 1-\rho\_2$, indicating a decrease, this reduction is linear with respect to the number of iterations (as suggested by the term $1-T\rho$ in Proposition 2). However, the number of models generated by RashomonGB grows exponentially with the number of iterations (Line 158). Thus, despite the decreasing probability, the total number of models in the Rashomon set ($m^{T\_2} \times (1-\rho\_2)$) will asymptotically increase with the number of iterations.
> We hope the explanation reduces the confusion, and will add the clarification in the revised version.
>
>
> We agree with the reviewer's comment that the size of the "empirical" Rashomon set (see Line 82) is not a proxy for multiplicity.
> For instance, an empirical Rashomon set with 100 globally diverse (e.g., obtained by re-training with different seeds) models might exhibit a higher predictive multiplicity metric (e.g., VPR) compared to another empirical Rashomon set containing 1000 models that differ only locally. The size of the "true" Rashomon set (see equation (1)), on the other hand, representing an ideal scenario achievable with unlimited computational and storage resources, can indeed act as a proxy for predictive multiplicity. In this context, predictive multiplicity metrics are non-decreasing with a larger size of the true Rashomon set (i.e., a larger $\epsilon$).
> We are grateful to the reviewer for highlighting the ambiguity in our discussion. We will clarify the distinction between the true and empirical Rashomon sets in Section 3 and in the sections discussing empirical studies in the revised manuscript.
>
> For Question 3, we are thankful for the reviewer's feedback in the initial review round, which guided us in enhancing the presentation of Figure 3. We are glad that the reviewer acknowledges the clarity brought by the additional figure and explanation. In the revised version, we will ensure to include detailed explanations on how to interpret Figure 3 effectively.
>
> === changing the typo in $1-\rho\_1 \leq 1-\rho\_2$ to $1-\rho\_1 \geq 1-\rho\_2$ as pointed out by Reviewer JGsC in the discussion. ===

---

> > ### Comment · Reviewer_JGsC · 2024-08-12
> >
> > Your response helped me narrow down where my confusion came from. I had not considered the exponentially growing number of models themselves, which despite a decreasing probability of a model being in the Rashomon set, would still overall make a bigger Rashomon set. Thank you for the clarification.
> >
> > Small correction: I believe the authors meant to write $1-p_1 \geq 1-p_2$, and not the other way around. Probably a small typo.
> >
> > To summarize our discussion, make sure to fix equation 4 and add appropriate information on how to interpret the figures. As for my concerns and confusion with increasing epsilon, it might be an artifact of my own reading of the work and not necessarily anything missing in the paper, but I’d encourage the authors to add appropriate clarifications and incorporate a discussion of how things evolve under a fixed epsilon, which, in my opinion, is a more realistic setting.
> >
> > Good work!

---

> > > ### Author Response · Authors · 2024-08-12
> > > **Further response to Reviewer JGsC**
> > >
> > > We appreciate the reviewer for guiding us to improve the quality of this manuscript, and for the summary of our discussions. We have fixed the typo in our comment above and left a note. We would definitely include our discussion and the reviewer's suggestion in the revised version! Thanks again!

---

### Official Review · Reviewer_HS14 · 2024-07-13

**Soundness:** 3
**Presentation:** 3
**Contribution:** 3
**Rating:** 6
**Confidence:** 2

**Summary:**

The paper explores the concept of predictive multiplicity in gradient boosting models. The Rashomon effect refers to the existence of multiple models that perform similarly well on a given dataset. The authors formalize this effect in the context of gradient boosting, introduce a new method called RashomonGB to efficiently explore this multiplicity, and demonstrate its application on various datasets. The paper aims to improve the estimation of predictive multiplicity and model selection, especially with considerations for group fairness.

**Strengths:**

1. The introduction of RashomonGB represents a novel method for exploring the Rashomon set in gradient boosting, offering an exponential search space as opposed to traditional linear methods.
2. The paper provides a robust theoretical foundation using statistical learning and information theory to analyze the Rashomon effect, enhancing the understanding of this phenomenon in gradient boosting.
3. The authors demonstrate the practical utility of RashomonGB on a wide range of real-world datasets, including tabular and image data, showcasing its versatility and effectiveness.

**Weaknesses:**

1. While the paper discusses the positive societal impacts of RashomonGB, it lacks a thorough exploration of potential negative impacts or misuse of the method.
2. The theoretical analysis relies on several assumptions that may not hold in all practical scenarios, potentially limiting the generalizability of the findings.
3. The paper mentions the intention to release code post-review, but the lack of immediate open access to code and data can hinder reproducibility and independent validation by other researchers.
4. Implementing RashomonGB might be complex for practitioners without a strong background in the theoretical aspects of machine learning and gradient boosting, potentially limiting its adoption in the industry.

**Questions:**

1. Can the method be extended or adapted for other types of machine learning models beyond gradient boosting?
2. How does the choice of hyperparameters in RashomonGB affect the stability and reliability of the results?
3. What are the practical challenges faced during the implementation of RashomonGB, and how can they be addressed to facilitate broader adoption?

---

> ### Author Rebuttal · Authors · 2024-08-06
>
> We appreciate the reviewer's feedback and questions.
>
> For Weakness 1, in the Introduction (Lines 26-29), we discuss the beneficial aspects of the Rashomon effect within the framework of responsible machine learning, highlighting its role in fairness by imposing additional constraints on models. This can be seen as searching for a fair model within the Rashomon set, which is further elaborated in Section 4.2 and [19].
> On the other hand, the negative implications of the Rashomon effect (Lines 30-35), suggest that predictive multiplicity may occur, leading to decisions for certain individuals being arbitrarily based on randomness in the training process rather than on learned knowledge.
> Moreover, the RashomonGB, along with other methods designed to monitor predictive multiplicity, could lead to negative societal impacts. For example, it might be exploited by service providers to identify models within the Rashomon set that disadvantageously affect the benefits (e.g., loan approvals) of certain populations, without showing significant statistical differences from non-discriminatory models. This could contribute to what we term an "Algorithmic Leviathan" [21], where discrimination and bias are concealed under the guise of algorithmic arbitrariness.
>
> For Weakness 2, specifically, we treat the loss function as a random variable, and assume that this loss function behaves as a sub-Gaussian random variable. Sub-Gaussianity is a practical assumption, as it can be easily achieved by clipping the loss (Lines 184-188).
> It is also a common assumption in theoretical analysis as it effectively generalizes the concept of boundedness [71, 75]. Adopting the sub-Gaussian assumption enables us to conduct a more expansive analysis.
> Additionally, we would like to highlight the novelty of our approach in establishing dataset-related bounds on the Rashomon set (i.e., Proposition 1). To the best of our knowledge, previous studies on the Rashomon effect have primarily focused on characterizing the Rashomon set through its hypothesis space. Our approach provides a fresh perspective and a significant contribution to the understanding of the Rashomon effect and predictive multiplicity in machine learning. We will enhance the clarity of these points in the revisions to Section 3.2.
>
> For Weakness 3, we plan to release the code, but only after the review process is complete for two reasons: First, releasing the code during the review phase could compromise the anonymity required by the double-blind review process in the NeurIPS guidelines. Second, early release could potentially infringe upon the protection of our intellectual property. Despite not releasing the code at this stage, we have provided a detailed, step-by-step procedure in the paper on how to construct the RashomonGB and reproduce our findings, specifically in Figure 1, Sections 4, and Appendices B.3 and D (Lines 266-270). This should enable reviewers and readers to understand and evaluate our methodology thoroughly. Moreover, the datasets used in this paper are all public available, and detailed descriptions including preprocessing can be found in Appendix D.
>
> For Weakness 4, it's important to clarify that the RashomonGB method can be straightforwardly implemented by training multiple models (i.e., weak learners) at each iteration, selecting $m$ models that meet the loss constraints defined by the Rashomon set, contrary to training only one model as illustrated in Figure 1. Additionally, during the inference phase, RashomonGB employs the method expansion outlined in Section 3.1 and depicted in Figure B.6 of Appendix B.3. This adaptability makes RashomonGB particularly suitable for industry applications, especially given that tabular datasets are still prevalent. Furthermore, although the training and inference processes of RashomonGB are simple and user-friendly, the effectiveness of the approach is underpinned by rigorous and innovative propositions detailed in Section 3. These factors collectively ensure that RashomonGB is not only practical but also robust, enhancing its applicability in various real-world scenarios.
>
> For Question 1, yes, we demonstrate the utility of RashomonGB beyond beyond decision trees as weak learners by incorporating alternative weak learners, such as convolutional neural networks (shown in the results for CIFAR-10 in the lowest row of Figure 3 and the GrowNet in reference [6]), and linear regression (illustrated in Figure E.9 of Appendix E.4). This flexibility confirms that the methodology developed in our study is adaptable to a variety of settings beyond conventional gradient boosting.
>
> For Question 2, the key hyper-parameters of RashomonGB include the number of iterations $T$ and the number of models per iteration $m$, which ablation studies are included in Appendix E.6 and E.7. With a constant probability $\rho$ as per Proposition 3, increasing $T$ accumulates the Rashomon effect at each iteration, which in turn increases $\epsilon$.  Additionally, increasing the number of models $m$ per iteration allows RashomonGB to explore a broader range of models, thereby enhancing the reliability of model selection and the estimation of predictive multiplicity.
>
> For Question 3, current implementations of gradient boosting, such as those available in Scikit-Learn or XGBoost, do not support training multiple models within a single iteration. Additionally, RashomonGB requires a unique filtering process during each boosting iteration to regulate the loss deviation, which is critical for constructing the Rashomon sets. Please also refer to our reply for Weakness 4.
>
> Thanks again for the comment! We are happy to provide more information or answer any follow-up questions.

---

> > ### Comment · Reviewer_HS14 · 2024-08-14
> >
> > Thank you for your detailed reply. The rebuttal is very appreciated. I increased my score to 6.

---

### Author Rebuttal · Authors · 2024-08-06

We would like to thank the reviewers for their time and effort in reading and commenting on the manuscript. We appreciate that the reviewers found that the paper **“study a novel problem”** (Reviewer HS14, JGsC, XEPy, and E8Ec), **“has robust and interesting analysis on dataset-related Rashomon set bound”** (Reviewer HS14, JGsC and XEPy), and **"propose RashomonGB which is easy to implement and its practical utility validated with real-world datasets "** (Reviewer E8Ec and HS14). Below, we outline the specific enhancements and changes that will be incorporated into the revised version of our paper.

- We clarify the contributions of this paper. The Rashomon effect and predictive multiplicity are not yet studied for gradient boosting algorithms, which is widely used for tabular datasets, especially in the field of responsible machine learning. With empirical studies, we present both the positive (fair model selection in Section 4.2) and negative (predictive multiplicity in Section 4.1) impacts of the Rashomon effect for gradient boosting, along with two algorithms to mitigate predictive multiplicity (Section 4.3). The analysis we developed here is not limited to gradient boosting, i.e., decision tree weak learners, but can also be applied to other sequential learning schemes with neural network (GrowNet) or linear regression weak learners (CIFAR-10 results in Figure 3 and Figure E.9 of Appendix E.4).

- We clarify and add more discussions on the theoretical results regarding the dataset-related Rashomon set bound. We explain that the sub-Gaussian assumption for the loss random variable is a generalization of boundedness, and is a pratical and common assumption for information-theoretic bounds. Moreover, we explain that the Rashomon effect on the residual cumulates with the number of iterations and increase the Rashomon parameter under the same $\rho$. We also refer the reviewers to Figure E.8 in the Appendix, which demonstrates that fixing $\epsilon$ while
re-training with different random seeds results in a decreasing percentage of
models ($\rho$ from Proposition 1) in the Rashomon set. This observation corroborates
Proposition 3’s findings on the relationship between the mutual information, $\epsilon$ and $\rho$.

- We clarify the comparison between RashomonGB and Re-training for estimating predictive multiplicity metrics and fair model select, and how to properly interpret Figure 3 in the main text in Rebuttal Figure R.1. We first perform re-training with different random seeds and perform RashomonGB by using the same weak learners, i.e., the training cost of RashomonGB and Re-training are the same and hence the comparison presented in the paper is fair.

- We provide an additional experiments to compare RashomonGB with three baselines in estimating prediction uncertainty for gradient boosting regression trees, including the NGBoost [R1], PGBM [R2], and IBUG [R3] in Rebuttal Figure R.2. Note that prediction uncertainty aims to estimate $P(Y|x)$ given a sample $x$ while predictive multiplicity aims to construct a Rashomon set with multiple models with similar losses. As observed in Figure R.2, RashomonGB is able to cover more diverse models than NGBoost, PGBM, and IBUG, and hence less under-estimates predictive multiplicity metrics.


Finally, we would like to point out that the core problem in Rashomon effect research is to explore models in the Rashomon set. For a general hypothesis space it is computationally infeasible. Therefore, there is a fundamental trade-off between the efficiency and effectiveness of exploring diverse models in the Rashomon set. Re-training is the de facto method that gives the most diverse models in the Rashomon set while also the most inefficient. The RashomonGB trades in the diversity by searching for "local" models, in return of improving the efficiency.
Please feel free to follow up! We very much welcome further discussions.






[R1] Duan, Tony, Avati Anand, Daisy Yi Ding, Khanh K. Thai, Sanjay Basu, Andrew Ng, and Alejandro Schuler. "Ngboost: Natural gradient boosting for probabilistic prediction." In International conference on machine learning, 2020.

[R2] Sprangers, Olivier, Sebastian Schelter, and Maarten de Rijke. "Probabilistic gradient boosting machines for large-scale probabilistic regression." In Proceedings of the 27th ACM SIGKDD conference on knowledge discovery and data mining, 2021.

[R3] Brophy, Jonathan, and Daniel Lowd. "Instance-based uncertainty estimation for gradient-boosted regression trees." Advances in Neural Information Processing Systems, 2022.

---

### Decision · Program_Chairs · 2024-09-25

**Decision:**

Accept (poster)

**Comment:**

The authors introduce a novel approach to characterizing predictive multiplicity for gradient boosting. The paper also explores the implications for finding a group fair model with a sufficient level of accuracy. Reviewers appreciated the information theoretical analysis of the Rashomon effect in this setting, and saw a potential for impact of the proposed method given the prevalence of GB in tabular learning, along with the importance of fairness in tabular settings. Some concerns around the presentation and attention to detail seem to have been resolved during the rebuttal period.